# Introns control stochastic allele expression bias

Bryan Sands[1], Soo Yun[1] & Alexander R. Mendenhall ● [1✉]

Monoallelic expression (MAE) or extreme allele bias can account for incomplete penetrance, missing heritability and non-Mendelian diseases. In cancer, MAE is associated with shorter patient survival times and higher tumor grade. Prior studies showed that stochastic MAE is caused by stochastic epigenetic silencing, in a gene and tissue-specific manner. Here, we used *C. elegans* to study stochastic MAE in vivo. We found allele bias/MAE to be widespread within *C. elegans* tissues, presenting as a continuum from fully biallelic to MAE. We discovered that the presence of introns within alleles robustly decreases MAE. We determined that introns control MAE at distinct loci, in distinct cell types, with distinct promoters, and within distinct coding sequences, using a 5'-intron position-dependent mechanism. Bioinformatic analysis showed human intronless genes are significantly enriched for MAE. Our experimental evidence demonstrates a role for introns in regulating MAE, possibly explaining why some mutations within introns result in disease.

[1] Department of Laboratory Medicine and Pathology, School of Medicine, University of Washington, Seattle, WA, USA. ✉email: alexworm@uw.edu

Monoallelic expression can explain missing heritability, incomplete penetrance, non-Mendelian patterns of inheritance and manifestation of disease. This stochastic autosomal allele bias can manifest as a continuum of expression states, from minor allele expression imbalance to extreme bias or monoallelic expression. This kind of allele expression bias is not imprinting[1] or X-linked inactivation[2], detailed in Chess, 2016[3]. This form of stochastic autosomal allele bias has been observed by numerous independent groups over several decades, reviewed in Chess, 2016[3]. In 2007, using array hybridization technology, extreme allele expression bias was found to be widespread among human autosomal genes and verified with in situ hybridization[4]. Using ChIP-seq, RNA-seq, qPCR and in situ hybridization to assess allele expression bias, many additional studies confirmed the existence of widespread, extreme autosomal allele expression bias[3,5–15]. In some biological samples, extreme allele expression bias is less prevalent[16], leading to vigorous debate[17,18]. Yet, when surveying existing data for evidence of extreme bias in the cells of many different tissue types, a recent meta-analysis indicates that between 10 and 25% of genes can be expressed in an extremely biased or monoallelic fashion[19].

Stochastic autosomal allele bias can be the cause of differences in immune cell function, differences in manifestation of genetic diseases, and differences in cancer outcomes. Monoallelic expression has been reported in multiple immune cell types, thereby affecting immune cell function[20–25] (in addition to the distinct cases of B and T cell receptors, reviewed in ref. [3]). Monoallelic expression of alleles causes non-Mendelian patterns of both dominant and recessive genetic disease inheritance[26,27]. Extreme allele bias is likely causative in non-Mendelian patterns of inheritance for several other autosomal dominant genetic diseases, reviewed in ref. [8]. Finally, the role that stochastic allele bias plays in the development of cancer is becoming clear. For example, in patients that are heterozygous for mutations in *IDH1*, monoallelic expression is associated with shorter patient survival times and higher tumor grade[28]. Extreme allele expression bias of *BRCA1/2, DAPK1* or *APC* are risk factors for breast cancer[29], chronic lymphocytic leukemia[30] and colorectal cancer[31], respectively.

Stochastic autosomal allele bias is a conserved phenomenon among eukaryotes and metazoans. It has been observed independently in yeast[32,33], worms[34,35], flies[36], mice[6,37] and humans[4,5], with each group developing their own terminology for the phenomenon. Scientists and clinicians have referred to stochastic autosomal allele bias as allele-specific expression (ASE)[12,38–41], random autosomal monoallelic expression (RAME/RMAE)[3,37,42], monoallelic expression (MAE)[5,6,43,44], allelic imbalance[29], or allele differential expression (ADE)[33]. Scientists using fluorescent reporters to study expression bias in bacteria (which do not have alleles per se)[45] and yeast[32] referred to and measured the relative degree of bias as intrinsic noise. Here we will refer to the differences in allele expression ratio thought to be caused by stochastic autosomal allele silencing[3,7,8] as what we directly measure, stochastic allele bias. We will quantify stochastic allele bias as intrinsic noise[45]. Intrinsic noise measures the relative deviation from a 1:1 ratio of allele expression and is appropriate for quantifying and comparing the relative degrees of allele bias detected among cells in a tissue.

While it is known that partial or complete transcriptional silencing of alleles plays a significant role in causing allele bias[3,7,8], the molecular genetic mechanisms that control allele bias are not clear. Moreover, a lack of animal models wherein allele bias can be directly observed has hindered the study of MAE/bias in intact, biological systems[46]. The addition of genetically tractable animal models in which allele bias can be directly observed in live cells will help determine how stochastic allele bias manifests in vivo. Animal models for stochastic allele bias[34–36] will also be important in determining the mechanisms that control stochastic allele bias, in terms of initiation, maintenance and propagation, including associated consequences[8]. Understanding this phenomenon in metazoans is critical for improving health outcomes where allele bias is the cause of, or contributor to, disease.

We previously used *C. elegans* as a model to investigate non-genetic variation in gene expression[35,47,48]. Here, we used our methods for quantifying allele expression in vivo[34] to survey the extent of stochastic allele bias in *C. elegans* somatic tissues, and to identify *cis* factors that control it. We quantified the expression of alleles controlled by ubiquitous and tissue-specific promoters expressed from distinct loci in the *C. elegans* genome. This approach allowed us to directly observe how stochastic allele expression bias manifests in the cells of distinct tissues in a live, adult metazoan.

In this investigation we focused on introns as *cis* factors that might influence stochastic allele bias. Introns are noncoding cis DNA elements found within the coding sequence of most genes that can act as enhancers of gene expression and provide means for producing multiple gene products through alternative splicing, reviewed in refs. [49,50]. Introns have been bioinformatically[51] and experimentally[52] shown to increase active, open chromatin markings. Introns have also been shown to affect heritable, complete silencing of gene expression in the germline of *C. elegans*[53,54]. In single cell RNA-seq data, SNPs within introns correlated with a significantly higher probability of allele bias compared to SNPs in UTRs or exons[55]. These reports suggest introns might act as cis elements that prevent stochastic allele bias in somatic cells.

Here, we hypothesized that removing introns would result in increased allele bias in somatic cells. Accordingly, we removed introns from reporter alleles and natural gene sequences and found that, in most cases, the loss of introns resulted in an increase in allele bias, including more MAE. Thus, we uncovered a new role for introns in controlling MAE. This approach also allowed us to gain new insights into how introns, promoters, cell types and locus affect stochastic allele bias.

## Results

**Surveying stochastic allele bias across tissues**. To identify tissues with detectable stochastic allele bias at the protein level, we surveyed somatic tissues of animals expressing differently colored fluorescent *hsp-90* reporter alleles using a point scanning confocal microscope. Figure 1 shows the experimental design. Figure 1 details where we edited the genome (Fig. 1a), shows cartoon schematics of fluorescent alleles with and without introns (Fig. 1b), details how we quantify fluorescent alleles in vivo (Fig. 1c–e), and shows what *C. elegans* with high- and low-allele bias in their intestine cells could look like (Fig. 1e). Supplementary Fig. 2 shows what animals with high- and low-allele bias in their intestine cells would look like in color-blind-accessible blue and red.

We chose to start our investigation with *hsp-90* for a few reasons. First, reporter alleles of *hsp-90* are constitutively, ubiquitously expressed in somatic cells at a level that is readily quantifiable via confocal light microscopy in *C. elegans*[34,35]. Second, *HSP90* has a role in the development and progression of cancer[56–64]. Third, *HSP90* is also a conserved capacitor of phenotypic variation across species, from plants[65] to invertebrates[66–68] to vertebrates (zebrafish)[69]. Finally, *HSP90* was listed as monoallelically expressed in the monoallelic expression database (dbMAE)[43]. If MAE for *HSP90* is a conserved phenomenon between worms and humans, we

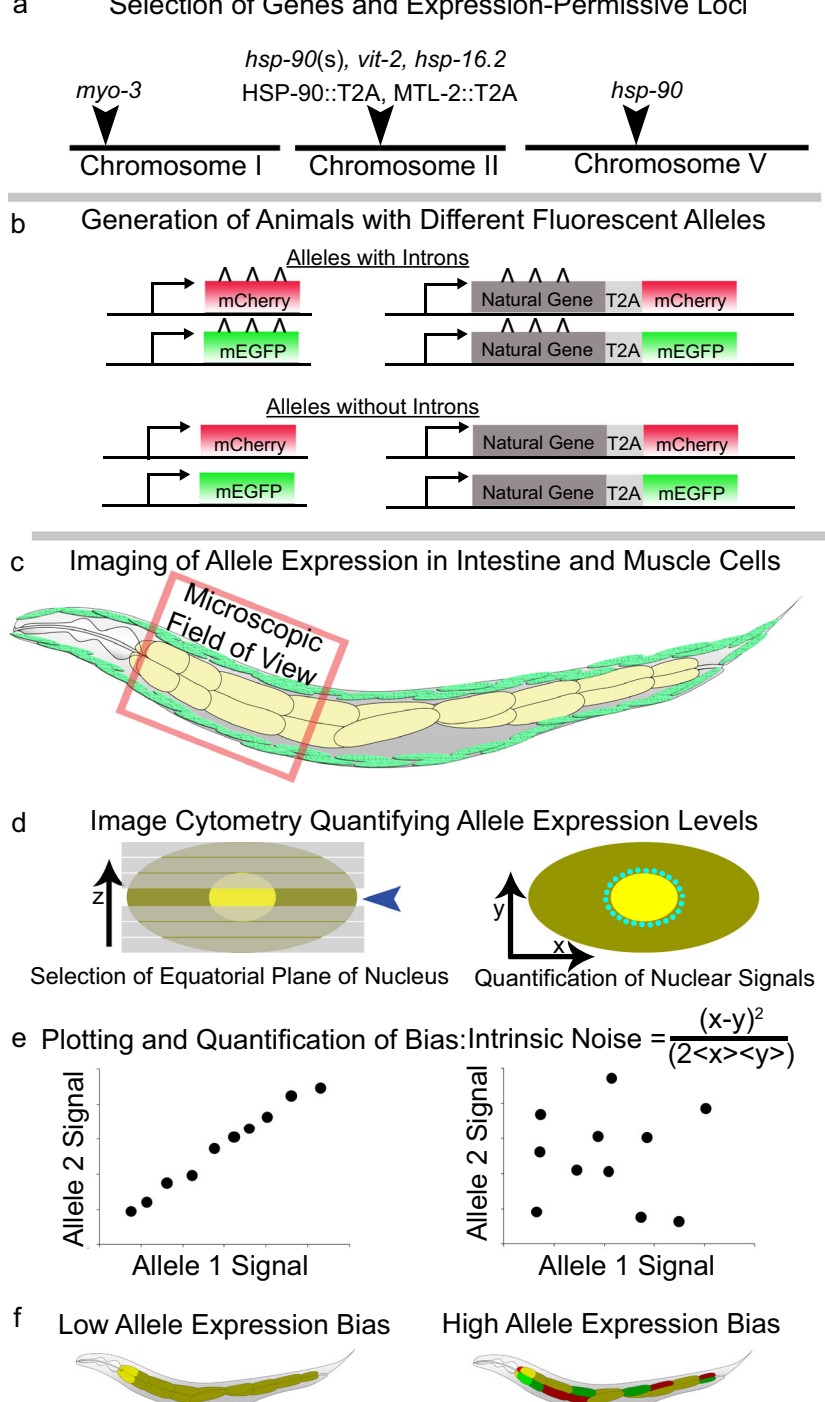

**a  Selection of Genes and Expression-Permissive Loci**

**b  Generation of Animals with Different Fluorescent Alleles**

**c  Imaging of Allele Expression in Intestine and Muscle Cells**

**d  Image Cytometry Quantifying Allele Expression Levels**

Selection of Equatorial Plane of Nucleus    Quantification of Nuclear Signals

**e  Plotting and Quantification of Bias:** Intrinsic Noise $= \dfrac{(x-y)^2}{(2\langle x\rangle\langle y\rangle)}$

**f  Low Allele Expression Bias        High Allele Expression Bias**

hypothesized we would be able to detect bias in at least one tissue, as per the requirements for entry into the dbMAE for mammals.

When we surveyed allele expression bias in somatic tissues, we found that strong allele bias was fairly prevalent, especially in diploid tissues, shown in Supplementary Fig. 1. In most cells, stochastic allele bias presented as a continuum among the cells in a tissue, with some cells manifesting monoallelic expression. We were able to visually detect strong allele bias in the cells of several distinct tissues including striated muscle cells, intestine cells, a dorsal nerve cord, the excretory cell, smooth muscle cells of the pharynx, and arcade cells (Supplementary Fig. 1b–h). Based on the observation of stochastic allele bias, and the practicality of measuring expression in different cell types based on size,

number, microscopic accessibility and occlusion-free signal, we determined that striated muscle cells and intestine cells would most suitably allow us to determine the role of introns in controlling stochastic allele bias.

**Effects of introns on stochastic allele bias in muscle cells**. To test if introns affected allele bias, we quantified expression of sets of *hsp-90* reporter alleles, with and without introns, in muscle cells, shown schematically in Fig. 1. Alleles were expressed from an autosomal locus on chromosome II, shown in Fig. 1a. For intron-bearing alleles, we used three canonical introns typically used in *C. elegans* transgenes[70], shown in Fig. 1b. We found that

**Fig. 1 Experimental design. a** shows autosomal chromosomes, expression-permissive loci we selected and the genes we analyzed at each locus. We examined multiple versions of *hsp-90* alleles at chromosome II, detailed in Supplementary Table 1 and Supplementary Tables 2-3. **b** shows cartoon images of reporter alleles with and without introns. Each experimental inquiry requires the generation of four distinct reporter alleles. Left panel shows reporters where promoters control fluorescent alleles with or without introns. Right panel shows natural coding sequences with or without natural introns; a fluorescent protein is made each time each allele is translated. **c** shows the microscopic field of view we utilize to optically section animals with a confocal microscope. We use these images to extract allele expression levels from intestine and/or muscle cells. **d** shows how we quantify gene expression from optical section images of animals' torso section highlighted in **c**. Left panel shows how we choose the z-slice containing the equatorial plane of the nucleus to quantify gene expression from. Right panel shows how we quantify gene expression using the relatively pure fluorescent protein signal in the nucleus, taking the average voxel value as a measure of the concentration. Additional details available in image cytometry methods, and in the original methodological publication, Mendenhall et al. 2015. **e** shows an example of allele expression data plotted as a scatter plot. We plot the expression data for each allele from each cell using the allele expression values as the x,y coordinates for that cell. Left panel shows a scatter plot of what relatively low stochastic allele bias would look like. Right panel shows a scatter plot of what relatively high stochastic allele bias would look like. **f** Left panel shows a cartoon diagram of a worm with an intestine expressing a gene with red and green alleles with no stochastic allele bias, resulting yellow cells. Right panel shows a cartoon diagram of a worm with an intestine comprised of cells expressing a gene with significant stochastic allele bias, indicated by the mix of cells expressing just the red allele, just the green allele, or both alleles (yellow).

the presence of introns in alleles significantly decreased stochastic allele bias in muscle cells (Fig. 2a, b, $P < 0.001$). The scatter plot in Fig. 2a shows that allele bias presents as a continuum, ranging from virtually monoallelic to completely biallelic. Images of sections of muscle cells from animals expressing fluorescent *hsp-90* transcriptional reporter alleles with and without introns are shown in Fig. 3. Supplementary Table 1 lists median intrinsic noise measurements and Spearman $R^2$ for all alleles.

Next, we examined *myo-3* reporter alleles inserted on chromosome I. Unlike *hsp-90*, which is constitutively expressed in many tissues, alleles of *myo-3* are constitutively expressed solely in striated body wall muscles. Compared to the *hsp-90* alleles, the data for cells expressing each *myo-3* allele are less widely dispersed (i.e., less noisy) than for *hsp-90* (Fig. 2a–d). From the scatter plots in Fig. 2a, c, it's clear that muscle cells expressing *hsp-90* are more likely to show extreme bias and monoallelic expression compared to cells expressing *myo-3* promoter driven alleles. Moreover, *myo-3* alleles are expressed at a relatively lower level than *hsp-90* alleles, yet the presence of introns within *myo-3* alleles still significantly decreased intrinsic noise by 71% (Fig. 2c, d, $P < 0.001$). These results demonstrate that introns reduce the probability of stochastic allele bias under the control of two distinctly regulated promoters.

In the two preceding experiments, we held coding sequence and 3'UTR as constants, but the *myo-3* and *hsp-90*-controlled alleles were located at different loci in the genome. To test if introns would decrease the probability of bias for the same alleles at different loci, we moved the *hsp-90* alleles to autosomal chromosome V, an autosomal expression-permissive locus with a distinct chromatin signature[71,72]. Reporter alleles with introns expressed from chromosome V significantly reduced intrinsic noise (Fig. 2e, f and Supplementary Table 1, $P = 0.047$). Our results indicate that in muscles cells, introns within otherwise identical alleles decrease the chance that allele expression imbalance will occur. Introns significantly decrease the probability of partial or complete stochastic allele bias whether the alleles are controlled by a ubiquitous or tissue-specific promoter, and even if the same set of *hsp-90*-controlled alleles are moved to a distinct autosomal locus. See Table 1 for a condensed table of effects of introns on stochastic allele bias in muscle cells, or Supplementary Table 1 for a more detailed comparison. Strain details are shown in Supplementary Tables 2 and 3.

**Effects of introns on stochastic allele bias in intestine cells.** To test if introns affect allele bias in distinct cell types, we measured alleles with and without introns in the relatively large, polyploid intestine cells. As in muscles, *hsp-90* reporter alleles with introns decreased allele bias by over 90% (Fig. 4a, b, $P < 0.001$). Images of

intestine cells expressing *hsp-90* alleles with and without introns are shown in Fig. 5. To further test the robustness of the intron effect on allele bias in intestine cells, we tested two additional, distinctly regulated promoters, holding the chromosome II locus as a constant. First, we tested a ubiquitously expressed heat shock inducible promoter from the gene, *hsp-16.2*. We found that in heat shocked animals expressing *hsp-16.2* reporter alleles, introns decrease intrinsic noise by 62% (Fig. 4c, d, $P < 0.001$). Next we tested alleles under control of the intestine-specific *vit-2* promoter, which normally controls yolk protein production during adulthood. When allele expression is controlled by the *vit-2* promoter, introns significantly decrease intrinsic noise by 82% (Fig. 4e, f, $P < 0.001$).

Finally, to more robustly determine if introns can decrease allele bias in intestine cells, we moved the *hsp-90* alleles from the chromosome II locus to the chromosome V locus. At the locus on chromosome V, we found introns caused a significant decrease in stochastic allele expression bias (Fig. 4g, h, $P = 0.001$). These data show that introns significantly decrease the probability of stochastic allele bias under control of multiple, distinctly regulated promoters, and at two distinct autosomal loci. See Table 1 for a condensed table of effects of introns on stochastic allele bias in intestine cells, or Supplementary Table 1 for a more detailed comparison.

**Effects of locus, cell type and promoter on stochastic allele bias.** Locus has been shown to affect gene expression levels and patterns[47,53,72]. As we could hold cis elements constant and measure identical reporter alleles at distinct loci, we could analyze our data as a function of locus. For *hsp-90* reporter alleles, intrinsic noise on chromosome V was an order of magnitude higher than for the locus on chromosome II (Fig. 6a, $P < 0.001$). These data suggest that location within the genome can determine the allele bias setpoint. Yet, at high and low noise set points, introns within reporter alleles decrease allele bias compared to alleles lacking introns (Fig. 4).

As *hsp-90* alleles are expressed in both striated muscles and intestine cells, we were able to test if cell type had an effect on allele bias. In *C. elegans*, muscles cells are diploid and intestine cells are polyploid, with most cells being binucleate (32 N or 64 N). We hypothesized that polyploid tissues should be less noisy than diploid tissues. In fact, our *hsp-90* reporter alleles in intestine cells showed that indeed, polyploid intestines are less noisy overall than diploid muscles (Fig. 6b, $P < 0.001$), though other factors besides ploidy may also be at play.

Finally, because locus, coding sequence, 3'UTR and cell type were held constant for *vit-2*, *hsp-16.2* and *hsp-90* reporter alleles, we were able to analyze the role of promoter in stochastic allele

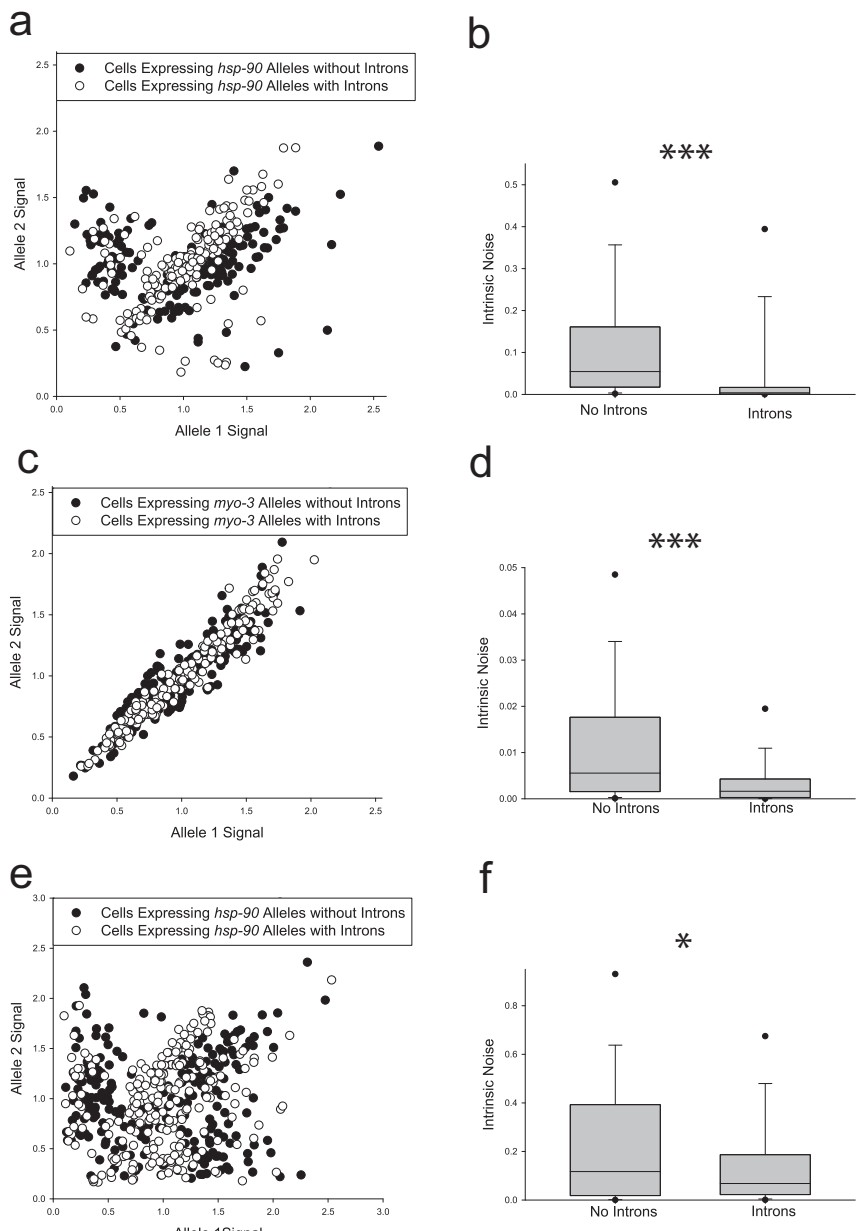

**Fig. 2 Stochastic allele bias in muscle cells.** Scatter plots show normalized allele expression data for individual cells expressing reporter alleles with and without introns. Boxplots show intrinsic noise measured from each cell for each set of reporter alleles with and without introns. Top of boxplot is 75th percentile, bottom of box is 25th percentile, line is median, top and bottom error bars are 90th and 10th percentile, respectively, and dots are 95th and 5th percentile. **a** shows a scatter plot of allele expression data from muscle cells expressing fluorescent alleles with and without introns, controlled by the *hsp-90* promoter. *n* = 180 cells per group examined over three independent experiments. **b** shows boxplots of intrinsic noise for each set of alleles in muscle cells from **a**. **c** shows a scatter plot of allele expression data from muscle cells expressing fluorescent alleles with and without introns, controlled by the *myo-3* promoter. *n* = 240 cells per group examined over four independent experiments. **d** shows boxplots of intrinsic noise for each set of alleles in muscle cells from **c**. **e** shows a scatter plot of allele expression data from muscle cells expressing fluorescent alleles with and without introns, controlled by the *hsp-90* promoter, expressed from a locus on chromosome V instead of chromosome II. *n* = 180 cells per group examined over three independent experiments. **f** shows boxplots of intrinsic noise for each set of alleles in muscle cells from **e**. Statistics: **b**, **d**, **f** Mann–Whitney two-tailed non-parametric test. *p < 0.05, **p < 0.01, ***p < 0.001. Source data are available in the Source Data file.

bias. Our analysis of all alleles sorted by promoter revealed that promoters had a significant effect on stochastic allele bias (Fig. 6c, P < 0.01 for all comparisons except *hsp-16.2* vs. *hsp-90* where P < 0.05). Taken together, our data show that allele bias is a complex phenomenon with genomic location, promoter, and cell type each contributing to an overall setpoint of allele bias. Yet, regardless of the loci, promoters and cell types we tested, introns still affected stochastic allele bias (Figs. 2 and 4).

**Effects of intron sequences and positions on stochastic allele bias.** In all of the above experiments, the intron-bearing alleles contained three synthetic introns that are commonly found in *C. elegans* transgenes[70]. In all scenarios tested, we found that alleles with introns significantly decreased stochastic allele bias. To test if natural intron sequences had similar effects on bias, we replaced the three synthetic introns in our mCherry reporter allele with two natural introns that occur in *hsp-90*. We matched the

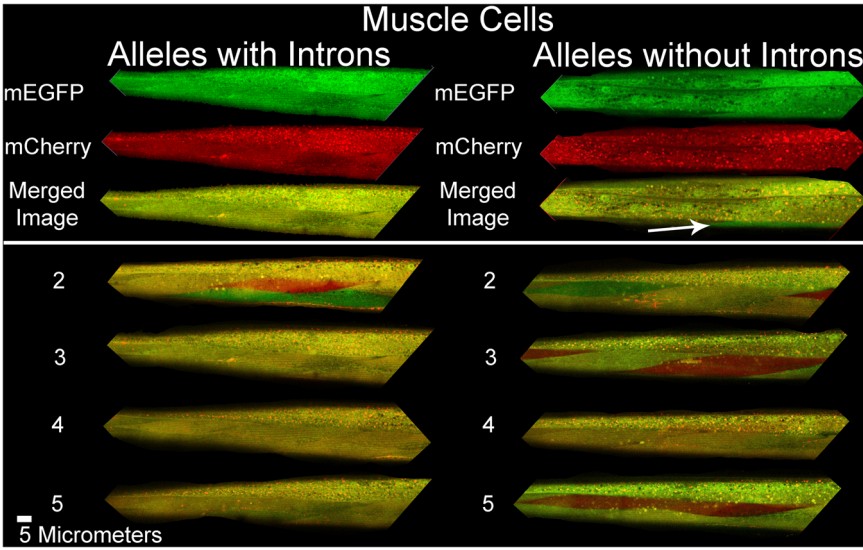

**Fig. 3 Micrographs of stochastic allele bias in striated muscle cells.** Composite images show merged red/green signal micrographs of animals expressing *hsp-90* reporter alleles in their striated body wall muscles. Top panel shows the individual red signal, green signal and merged micrographs. Left column shows sections of animals with muscle cells expressing alleles with introns. Right column shows sections of animals with muscle cells expressing alleles without introns. We arbitrarily selected five images per group from the z stacks from the three independent experiments used to generate the data in Fig. 2a.

**Table 1 Effects of introns on stochastic allele bias in muscle cells.**

| Cell type | Locus | Promoter or protein | Intron effect on allele bias |
|---|---|---|---|
| Striated muscle | II | *hsp-90* | YES |
| Striated muscle | V | *hsp-90* | YES |
| Striated muscle | I | *myo-3* | YES |
| Intestine | II | *hsp-90* | YES |
| Intestine | V | *hsp-90* | YES |
| Intestine | II | *hsp-16.2* | YES |
| Intestine | II | *vit-2* | YES |
| Intestine | II | HSP-90 | YES |
| Intestine | II | MTL-2 | NO |

position and splice junctions such that only the internal intron sequences were different (for intron sequence details see Supplementary Table 4). We found that even when one allele has three synthetic introns and the other allele has two natural *hsp-90* introns, stochastic allele bias was significantly reduced compared to intronless alleles, shown in Fig. 7a ($P < 0.001$). The 87% decrease in median noise caused by the natural/synthetic introns is similar to our results with all synthetic introns (93% decrease).

Intron positions can affect gene expression levels[50]. In *C. elegans*, a single 5'-intron is sufficient to increase gene expression level[70]. Evidence from cell culture experiments showed that 5'-introns increase the proportion of active, open chromatin markings, consistent with the hypothesis that 5'-introns would prevent stochastic allele bias[52]. Therefore, we placed a single intron in a relatively 5' position or a relatively 3' position within the coding sequence of each differently colored allele controlled by the *hsp-90* promoter and tested for effects on stochastic allele bias in intestine cells. Alleles with just a 5' intron had Spearman $R^2$ similar to alleles with 3 introns (87% and 90% $R^2$, respectively) and alleles with a 3' intron only had Spearman $R^2$ similar to intronless alleles (63% and 56% $R^2$, respectively, see Supplementary Table 1). When we compared intrinsic noise between 5' only,

and 3' only, we found alleles with a 5' intron significantly decrease the probability of stochastic allele bias compared to alleles with 3' intron, shown in Fig. 7c ($P < 0.001$).

**Effects of introns in distinct coding sequences on stochastic allele bias.** In the above experiments, we determined introns affect allele bias in different tissues, at different loci and under the control of distinct promoters. In all cases, we measured alleles containing introns in the context of the coding sequence of mCherry or mEGFP. To determine the effect of introns on allele bias in the context of natural genes with naturally occurring introns, we adopted a T2A approach. For these experiments, full length MTL-2 or HSP-90 coding sequences were fused to GFP and mCherry coding sequences using T2A peptides. T2A peptides are widely used ribosomal skip elements that allow for two or more proteins to be made from a single mRNA[73]. In worms, T2A peptides allow for a 1:1 ratio of the two expressed proteins[74]. In addition to *hsp-90*, we chose to also examine *mtl-2* because it contains a single, small, 5'-intron, is intestinally expressed, and because of the biological significance of metallothioneine proteins in stress response and cancer[75–77]. When we removed the natural introns from the coding sequence of *hsp-90*, HSP-90 was expressed in a more biased fashion (Fig. 7e, f, $P < 0.001$). However, when we removed the sole natural intron from *mtl-2*, MTL-2 was not expressed in a significantly more biased fashion (Fig. 7g, h, $P > 0.05$). Additional details are in Supplementary Table 1. These results demonstrate that, for *hsp-90* promoter-controlled genes, introns significantly decrease allele bias whether the coding sequence is a fluorescent protein or the HSP-90 chaperone coding sequence. However, the *mtl-2* result shows that not all introns in all genes have robust effects on allele bias.

**Bioinformatic analyses of intronless genes.** Our experiments showed that introns within alleles restrict allele bias and mono-allelic expression. Conversely, the intronless nature of a gene might then promote allele bias. If intronless genes provide a means to generate variegated allele expression by promoting allele bias towards one parental allele, we reasoned that certain genes might be selected for or against. If allele bias were selected for,

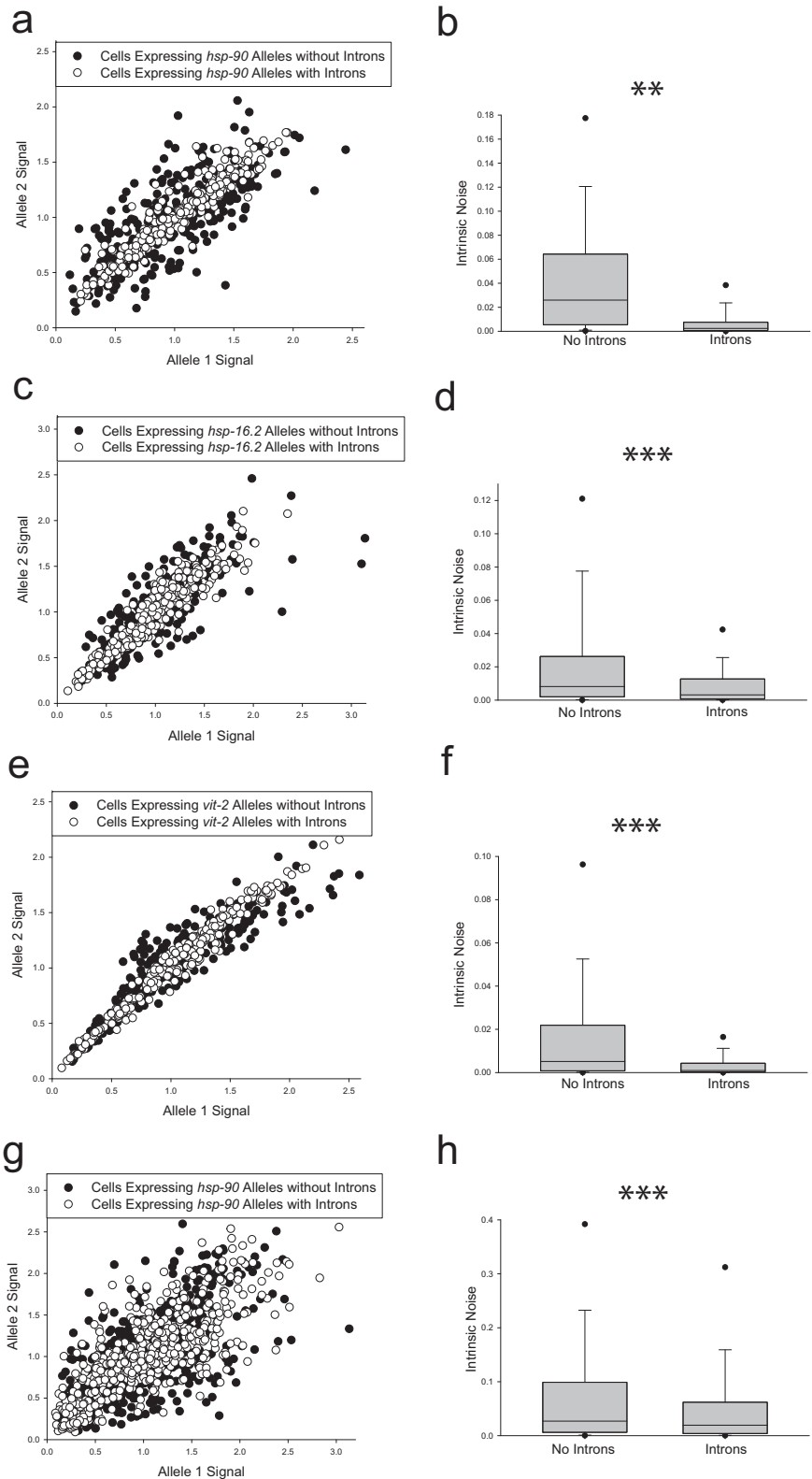

and if introns prevent allele bias, then intronless genes should be enriched for MAE or extreme allele bias. Of the 20,390 protein-coding genes in the human genome (GRCh38), 1164 are intronless—about 6% (Supplementary Data 1). When we compared the list of human intronless genes to dbMAE[43], we found that 64% of them are listed as monoalleleic. This is significantly

higher than the expected rate of 10–25% for all protein-coding genes (Supplementary Data 1).

We hypothesized that intronless genes may be enriched for specific molecular functions or biological processes. If there was enrichment for GO terms, it could suggest processes or functions for which allele bias might be beneficial (among other reasons for

**Fig. 4 Stochastic allele bias in intestine cells.** Scatter plots show normalized allele expression data for individual cells expressing reporter alleles with and without introns. Boxplots show intrinsic noise measured from each cell for each set of reporter alleles with and without introns. Top of boxplot is 75th percentile, bottom of box is 25th percentile, line is median, top and bottom error bars are 90th and 10th percentile, respectively, and dots are 95th and 5th percentile. **a** shows a scatter plot of allele expression data from intestine cells expressing fluorescent alleles with and without introns, controlled by the *hsp-90* promoter. $n = 278$ cells for the intronless group, and $n = 277$ cells for the introns group examined over four independent experiments. **b** shows boxplots of intrinsic noise for each set of alleles in intestine cells from **a**. **c** shows a scatter plot of allele expression data from intestine cells expressing fluorescent alleles with and without introns, controlled by the *hsp-16.2* promoter. $n = 277$ cells for the intronless group, and $n = 280$ cells for the introns group examined over four independent experiments. **d** shows boxplots of intrinsic noise for each set of alleles in intestine cells from **c**. **e** shows a scatter plot of allele expression data from intestine cells expressing fluorescent alleles with and without introns, controlled by the *vit-2* promoter. $n = 280$ cells for the intronless group, and $n = 279$ cells for the introns group examined over four independent experiments. **f** shows boxplots of intrinsic noise for each set of alleles in intestine cells from **e**. **g** shows a scatter plot of allele expression data from intestine cells expressing fluorescent alleles with and without introns, controlled by the *hsp-90* promoter expressed from a locus on chromosome V instead of chromosome II. $n = 559$ cells for the intronless group, and $n = 560$ cells for the introns group examined over eight independent experiments. **h** shows boxplots of intrinsic noise for each set of alleles in intestine cells from **g**. Statistics: **b** Kruskal–Wallis One Way Analysis of Variance on Ranks. Multiple comparisons: Dunn's Method. **d**, **f**, **h** Mann–Whitney two-tailed non-parametric test. \*$P < 0.05$, \*\* $P < 0.01$, \*\*\* $P < 0.001$. Source data are available in the Source Data file.

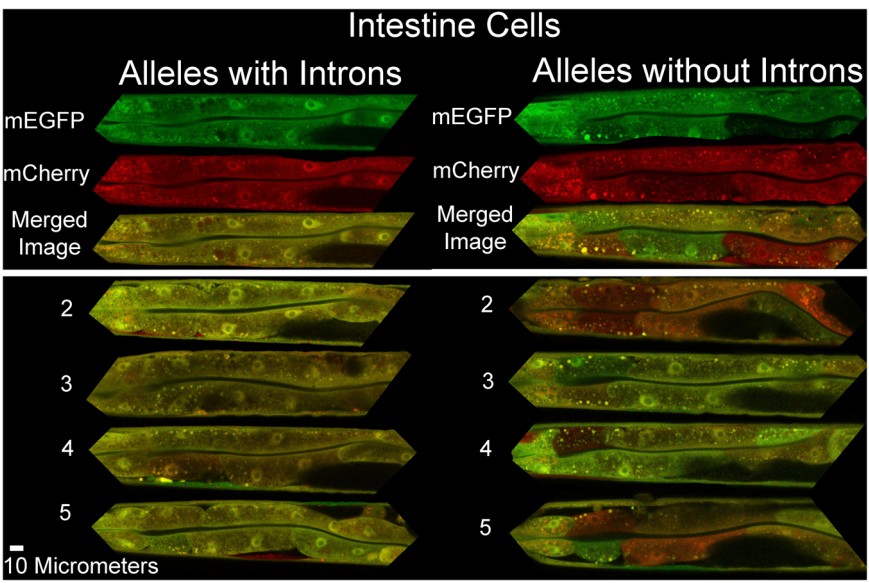

**Fig. 5 Micrographs of stochastic allele bias in intestine cells.** Composite images show merged red/green signal micrographs of animals expressing *hsp-90* reporter alleles in their intestines. Top panel shows the individual red signal, green signal and merged micrographs. Left column shows sections of animals with intestine cells expressing alleles with introns. Right column shows sections of animals with intestine cells expressing alleles without introns. We arbitrarily selected five images per group from the z stacks from the four independent experiments used to generate the data in Fig. 4a.

being intronless). We performed GO terms enrichment analyses on worm and human intronless genes for molecular functions and biological processes (Supplementary Data File 1). About 3% of *C. elegans'* protein-coding genes are intronless (2.6%, 529 genes). We found significant enrichment for dozens of functions and processes in both worms and humans. We found significant enrichment of *C. elegans'* intronless genes for 19 molecular functions and 21 biological processes. The top five distinct molecular functions for *C. elegans'* intronless genes were: protein heterodimerization activity, DNA binding, histone binding, NADH dehydrogenase (ubiquinone) activity, and protein-containing complex binding. The top five distinct biological processes for *C. elegans'* intronless genes were: nucleosome assembly, chromatin silencing, DNA repair, mitochondrial electron transport, and 3'-UTR-mediated mRNA destabilization.

In humans, we found significant enrichment of intronless genes for 64 molecular functions and 51 biological processes. The top five distinct molecular functions for human intronless genes were: G protein-coupled receptor activity, olfactory receptor activity, protein heterodimerization activity, type I interferon receptor binding, and nucleosomal DNA binding. The top five distinct biological processes for human intronless genes were:

detection of chemical stimulus, G protein-coupled receptor signaling, keratinization, nucleosome assembly, and chromatin silencing at rDNA. As expected, we found enrichment for "olfactory receptor activity". Olfactory receptors are known to be expressed in an exclusively monoallelic fashion[78], validating that the approach detects GO terms associated with MAE (Supplementary Data 1).

## Discussion

We have developed *C. elegans* as a model for studying allele bias in vivo. Using this system, we found that introns play a significant role in determining allele bias. We showed that this is true in diploid muscles, in polyploid intestine cells, at distinct loci, under control of distinct promoters, and in the context of three distinct coding sequences. We found that the position of the intron needs to be near the 5' region of the coding sequence to control bias. Moreover, intronless human genes appear to be overrepresented in a database of monoallelically expressed genes. Taken together, our data point to a complex regulatory environment where genomic locus and cis factors in genes determine a setpoint for allele bias, and demonstrate a new role for introns in controlling MAE. This study provides experimental evidence showing introns

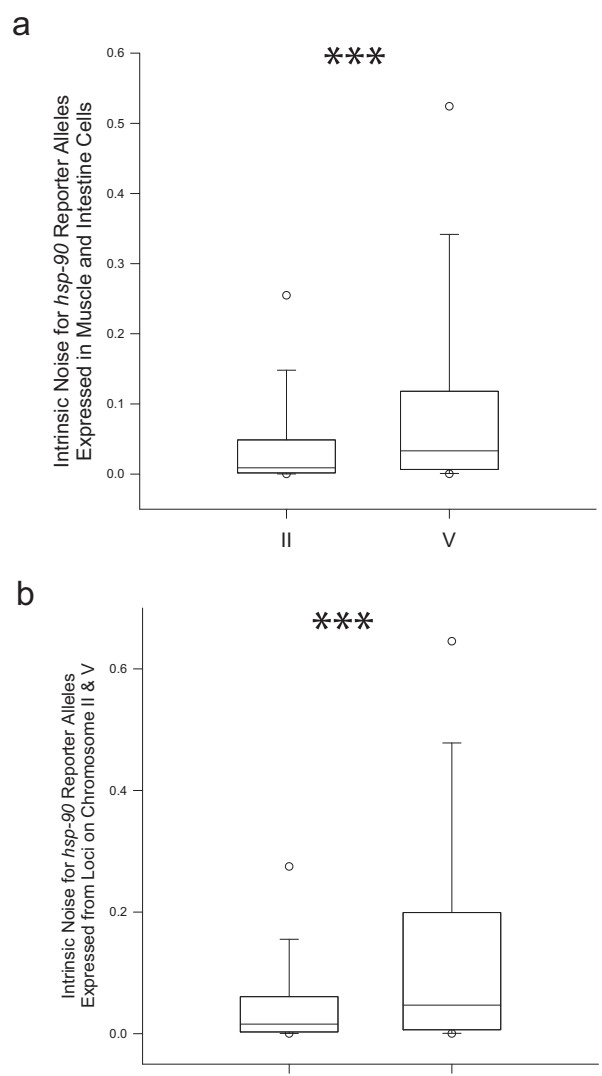

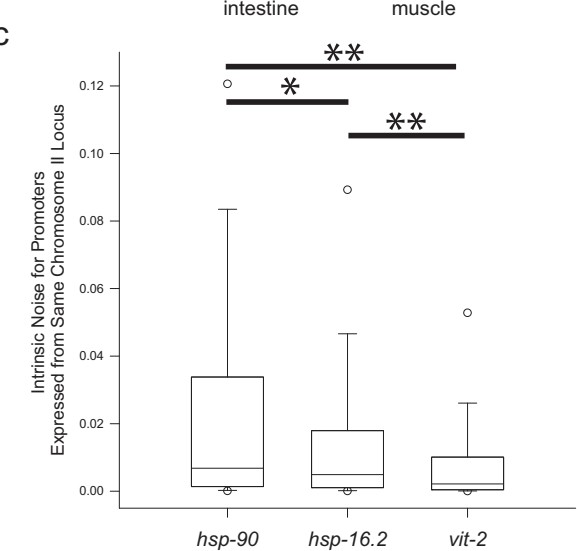

**Fig. 6 Locus, cell type, and promoter effects on noise.** Boxplots show intrinsic noise measured from each cell for each set of reporter alleles with and without introns, grouped by locus, cell type or promoter. Top of boxplot is 75th percentile, bottom of box is 25th percentile, line is median, top and bottom error bars are 90th and 10th percentile, respectively, and dots are 95th and 5th percentile. **a** shows boxplots of intrinsic noise for all cells expressing *hsp-90* reporter alleles grouped by locus. $n = 915$ cells for the chromosome II group, and $n = 1599$ cells for the chromosome V group analyzed from nineteen independent experiments. **b** shows boxplots of intrinsic noise for all cells expressing *hsp-90* reporter alleles in grouped by cell type. $n = 1674$ cells for the intestine group, and $n = 840$ cells for the muscle group analyzed from nineteen independent experiments. **c** shows boxplots of intrinsic noise for intestine cells expressing *hsp-90*, *hsp-16.2* or *vit-2* alleles from the chromosome II locus. $n = 555$ cells for the *hsp-90* group, $n = 557$ cells for the *hsp-16.2* group, and $n = 559$ cells for the *vit-2* group analyzed from twelve independent experiments. Statistics: **a**, **b** Mann–Whitney two-tailed non-parametric test. **c** Kruskal–Wallis One Way Analysis of Variance on Ranks. Multiple comparisons: Dunn's Method. *$P < 0.05$, ** $P < 0.01$, *** $P < 0.001$. Source data are available in the Source Data file.

bias. Previous investigations into monoallelic expression/extreme allele bias came from work with tissue or blood samples, and cells in culture. These studies found significant clinical implications for extreme allele expression bias (e.g., patient survival times in[28]), that it was widespread[4–6], and that it was associated with and/or controlled by chromatin markings[5,6,14,15]. These studies have yielded valuable clinical and scientific insights.

Animal models for monoallelic expression were lacking before 2015[34]. Microscopically accessible, genetically malleable, small animal model systems will contribute to the elucidation of molecular genetic control mechanisms of expression bias. These systems allow scientists to see the patterns of allele expression bias in vivo. A major question in the field is determining what controls the initiation, propagation and maintenance of allele bias[8]. For this conundrum, the *C. elegans* intestine may be ideal. The intestine starts as a diploid tissue with 20 cells in the hatched larvae, then undergoes endoreduplications in all cells, and nuclear divisions in some. The advantage here is that the same cells must initiate, propagate and maintain allele bias throughout development in the same cells, eliminating the need to track dividing cells. Furthermore, the observation of extreme allele bias in adult cells is highly improbable because they have 16 copies of each allele in each nucleus. This suggests that the initiation of allele bias occurs early in development, in the L1 larvae, and is propagated and maintained to result in the extreme bias we see in some adult intestine cells.

Stochastic allele bias may be best observed at the protein level in animal models. A *Drosophila* model investigating stochastic allele bias during development found that mRNA was not correlated with protein for individual alleles[36]. Moreover, mRNA has often not been well correlated with protein[79–81]. However, in cell culture, monoallelic expression of mRNA can be incredibly stable[4,14,15]. Cultured cells can maintain mitotically stable allele bias[3]. It is possible that the recent debate about the extent of monoallelic expression from freshly isolated cells may boil down to transcriptional bursting and adaptation[17,18]. Using fluorescent reporter alleles is not currently as high-throughput as RNAseq or ChIP-seq, but this complementary approach will continue to provide answers to important questions surrounding allele bias. Moreover, in vivo observations can be used to help interpret more global, transcript-based studies, as suggested in[6].

control stochastic allele bias. Figure 8 graphically summarizes the effects of introns on stochastic allele bias.

**Animal models of stochastic autosomal allele bias.** Animal models can significantly enhance the study of stochastic allele

**Introns, gene expression, and stochastic allele bias.** Recently, a study examining RNA-seq data found that mutations within

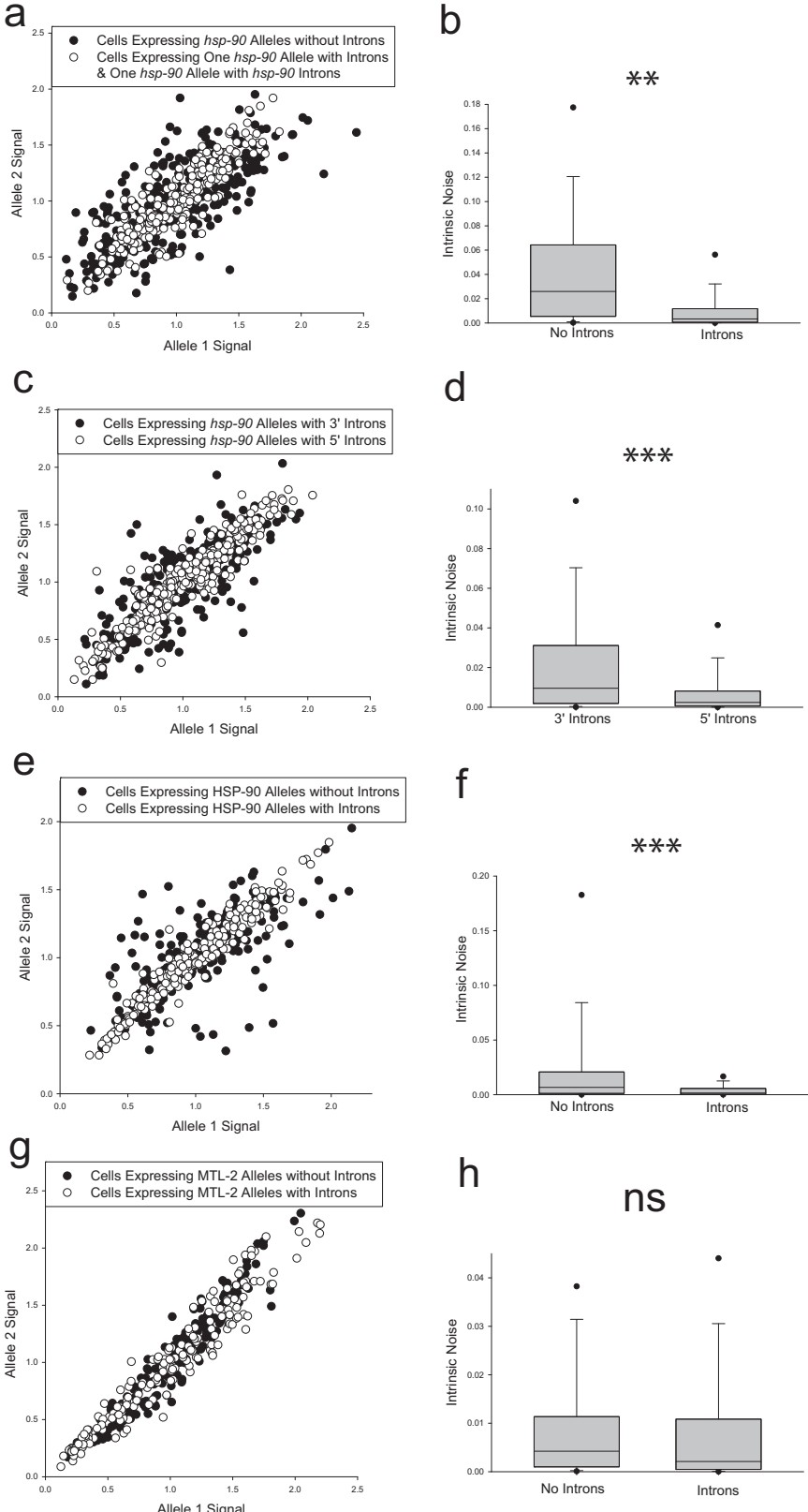

introns significantly decrease the likelihood of a gene being expressed, while mutations within 5' UTRs, 3' UTRs or exons did not show this effect[55]. In this study, it is possible that these point mutations are disrupting splicing, for example, by eliminating the branch point. However, our data suggests that another interpretation is possible—that point mutations in introns are

sufficient to negate the effect of introns on preventing allele bias via a loss of a cis element. Our experimental system can be used in future studies to directly determine the sequence requirements within introns that are necessary for the effect on allele bias. *C. elegans* has a much larger proportion of shorter introns compared to humans[70], with many introns that are less than 100 base pairs,

**Fig. 7 Stochastic allele bias in intestine cells with different introns sequences and positions.** Scatter plots show normalized allele expression data for individual cells expressing reporter alleles with and without introns, or with 5' or 3' introns. Boxplots show intrinsic noise measured from each cell for each set of reporter alleles. Top of boxplot is 75th percentile, bottom of box is 25th percentile, line is median, top and bottom error bars are 90th and 10th percentile, respectively, and dots are 95th and 5th percentile. **a** shows a scatter plot of allele expression data from intestine cells expressing fluorescent alleles with natural and synthetic introns, or without introns, controlled by the *hsp-90* promoter. n = 278 cells for the intronless group, and n = 279 cells for the introns group examined over four independent experiments. **b** shows boxplots of intrinsic noise for each set of alleles in intestine cells from **a**. **c** shows a scatter plot of allele expression data from intestine cells expressing fluorescent alleles with only 3' or 5' introns, controlled by the *hsp-90* promoter. n = 280 cells per group examined over four independent experiments. **d** shows boxplots of intrinsic noise for each set of alleles in intestine cells from **c**. **e** shows a scatter plot of allele expression data from intestine cells expressing HSP-90 alleles with and without introns, controlled by the *hsp-90* promoter. n = 252 cells for the intronless group, and n = 263 cells for the introns group examined over four independent experiments. **f** shows boxplots of intrinsic noise for each set of alleles in intestine cells from **e**. **g** shows a scatter plot of allele expression data from intestine cells expressing MTL-2 alleles with and without introns, controlled by the *mtl-2* promoter. n = 178 cells for the intronless group, and n = 179 cells for the introns group examined over three independent experiments. **h** shows boxplots of intrinsic noise for each set of alleles in intestine cells from **g**. Statistics: **b** Kruskal–Wallis One Way Analysis of Variance on Ranks. Multiple comparisons: Dunn's Method. **d**, **f**, **h** Mann–Whitney two-tailed non-parametric test. *P < 0.05, ** P < 0.01, *** P < 0.001. Source data are available in the Source Data file.

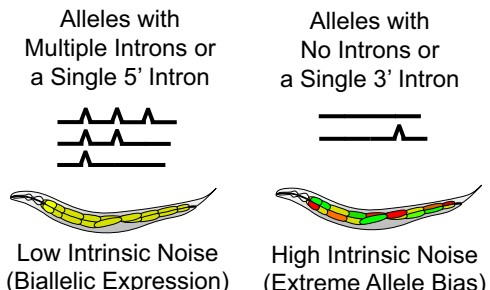

Alleles with Multiple Introns or a Single 5' Intron

Alleles with No Introns or a Single 3' Intron

Low Intrinsic Noise (Biallelic Expression)

High Intrinsic Noise (Extreme Allele Bias)

**Fig. 8 Summary of intron effects on stochastic allele bias.** Left panel shows intron configurations with lower stochastic allele bias. Right panel shows intron configurations with higher stochastic allele bias.

including the introns in *hsp-90* and *mtl-2* used in this study. Relatively short introns should be advantageous for identifying critical sequences preventing allele bias in future studies.

Here, we found that a single, 5'-positioned intron is sufficient to decrease the probability of stochastic allele bias. Put differently, a 5' positioned intron can promote biallelic expression. One of our previous studies found that a 5'-intron is sufficient for intron mediated enhancement of expression level in *C. elegans*[70]. Therefore, it seems reasonable to suggest that one of the mechanisms by which introns increase gene expression levels could be by preventing stochastic autosomal allele bias caused by silencing. Previous work has found that introns impart an active chromatin signature near the 5' region of a gene[51,52]. An active chromatin signature might decrease the probability of stochastic autosomal allele silencing. Regardless of the exact mechanism by which introns increase the probability that both alleles of a gene are expressed, simply increasing that probability will increase gene expression levels[49,50].

Previous reports found that the tissue a gene is expressed in can affect the amount of allele bias[6,19], and we confirmed that here. We also found that locus and promoter could affect allele bias. By moving identical reporter alleles from a chromosome II locus to a chromosome V locus, we could isolate the effect of locus on allele bias. We found the chromosome V locus to be nosier than the chromosome II locus. This makes sense because it has a distinct chromatin signature[71]. Despite the increase in noise of the chromosome V locus, we found that introns still decreased allele bias at this site. While it seems obvious that a promoter could affect allele bias, it now has strong experimental support.

We did not detect a difference in MTL-2 allele expression bias when the 5' intron was removed from the *mtl-2* coding sequence. There are two possible reasons for this. First, *mtl-2*, like other

metallothioneines, is an extremely short gene, with a coding sequence of only 303 nucleotides, and a first exon of only 16 nucleotides. Short gene sequences like those of the metallothioneins may be resistant to, or lack cis elements required for the epigenetic changes necessary for allele bias to occur. Second, unlike, *hsp-16.2*, the *mtl-2* gene has a constitutive expression level that allowed us to observe it without exogenous induction. Thus, an intriguing possibility is that allele bias could change in an intron-dependent fashion after induction of expression caused by a stressor, as MTL-2 is induced by exogenous heavy metals, like cadmium[75].

We found that intronless genes are overrepresented in dbMAE. Taken together with our experimental results, these data raise the possibility that the presence of introns in an allele could be under selective pressure for the effect on stochastic allele bias. Moreover, monoallelic expression could be an important source of phenotypic variation[19], and intronless genes, especially those involved in stress response and immunity, might benefit from altered physiological capacities caused by stochastic allele bias. The identification of intronless genes being enriched for immune-related biological processes and molecular functions is consistent with the idea that immune genes may have lost introns to cause MAE. MAE is presumably beneficial for the immune system, indicated by multiple reports of MAE in immune cells types[20–25].

**Stochastic allele bias, introns, and human disease**. Monoallelic expression caused by stochastic allele bias is associated with escape from genetic disease[26], and worse outcomes for cancer patients[28,30,31,82]. Additionally, some people harbor dominant oncogenes, but remain cancer free[83,84]; silencing of oncongenic alleles may be a reason. In the case of *PIT1*, fortunate monoallelic expression of a "good" allele protected some, but not all, family members from the negative consequences of a dominant *PIT1* allele[26]. In this study, the father and grandmother of the affected patient harbored the dominant allele, but no mRNA of that allele was detected. This case demonstrates MAE can cause non-Mendelian escape from disease caused by the dominant *PIT1* allele.

There are rare cases of neutral lipid storage disease with myopathy, where individuals are homozygous for a point mutation in an intron in PNPLA2 that is predicted to cause a splicing defect[85]. In this small number of patients with active disease, no PNPLA2 mRNA is detected. While the mechanism behind the lack of mRNA could be the production of highly unstable (and thus not detectable) mRNA, an alternative hypothesis is that the point mutation in the intron caused the loss of a cis sequence element that prevented silencing.

Finally, an intriguing clinical case of a single patient with COL6A2-associated Bethlem myopathy also suggests that mutations in introns can lead to allele bias and disease[27]. In this study, the affected patient harbored one COL6A2 allele with a large deletion in intron 1a, and a second COL6A2 allele with a six nucleotide deletion in exon 28 that is associated with disease in a recessive fashion. The authors showed that the allele with the deep intronic mutation was silenced, as the RNA was not detectably expressed. This resulted in sole expression of the disease allele (with the deletion in exon 28) and non-Mendelian manifestation of disease.

Our experimental data, the three aforementioned clinical reports, and the recent report showing mutations in introns correlate with allele bias[55], taken together, comprise a significant body of evidence suggesting that introns affect allele bias. Given this array of evidence, the idea that mutations in introns can affect human disease by affecting stochastic allele bias/MAE should be seriously considered whenever intronic mutations are associated with a disease. Extreme stochastic allele bias/mono-allelic expression can be consequential, might be a more prevalent cause of disease than originally thought, and we can now be quite certain that introns can affect this fundamental property of gene expression.

## Methods

**Molecular cloning and strain creation.** For MosSCI insertions[72,86,87], we generated all of the DNA constructs in BSP188 (Addgene110917) by 3-fragment DNA assembly in yeast, using a protocol that we recently published[88]. This expression vector contains the unc-54 terminator and chromosome II MosSCI homology arms for integration at ttTi5605. A list of primers used for assembly can be found in Supplementary Table 5. For promoter sequences, we used worm gDNA to amplify sequences upstream of the ATG as follows: 2Kb upstream of the ATG for hsp-90, 392 bp upstream for hsp-16.2, 4Kb upstream for vit-2, and 567 bp upstream for mtl-2. Intronless transgenes were assembled by overlap extension PCR using intron-containing transgenes as template. The T2A peptide sequence[73] was synthesized (IDT, Coralville, IA) and stitched to mtl-2 and hsp-90 gene fragments and reporter genes by overlap extension PCR before yeast assembly into vector BSP188. We rescued assembled DNA into E. coli and sequence verified the final assembled plasmids. Worm strains were generated by micro-injection of MosSCI or CRISPR repair templates. For CRISPR repair templates we used partially single-stranded PCR products as per Dokshin et al.[13] CRISPR edits were made in SKILODGE strains with additional information here[6]. We outcrossed each strain reported here with N2 wild-type animals a minimum of three times. The resulting strain names and genomic insertion designations are shown in Supplementary Table 2.

**Animal husbandry.** We maintained all strains in 10 cm petri dishes on NGM seeded with OP50 E. coli in an incubator at 20°. Additional details on animal culture conditions are available in ref. [89]. All strains used in this study are listed in Supplementary Table 2. A table of crosses can be found in Supplementary Table 3. To generate heterozygous GFP/mCherry expressing strains, we generated GFP expressing males by subjecting 20 L4s to a 30° heat shock for 5–6 h. For all strains, the GFP allele was introduced through the male germline. We screened for heterozygous animals that express both GFP and mCherry on a fluorescence stereoscope. We maintained heterozygotes by picking them away from homozygous animals and onto fresh, OP50-seeded NGM growth plates each generation. We performed experiments on heterozygous animals that were at least five generations beyond the initial cross to avoid paternal allele expression bias. To synchronize animals for experiments, we conducted 2 h egg lays onto 10 cm NGM plates (10 heterozygous animals per plate). For experiments with heat shock, we performed a one hour heat shock at 35° on one day old adult animals by placing animals on their NGM growth plates into a 35° incubator for one hour and then returning them back to the 20° incubator until imaging the next day, approximately 24 h later. All local, University and federal regulations regarding ethical invertebrate animal model research were followed.

**Microscopy.** We washed day two adult animals (second day of adulthood at 20°) into S-basal media with tricaine/tetramisole[34], and loaded animals in 80-lane microfluidic devices[88]. These devices immobilize worms in 80 separate lanes in a relatively restricted position, making presentation of the animals to the objective more uniform than using traditional agarose based slides. We imaged only those animals that randomly immobilized with their left side facing the cover slip, to which the fluidic device was bonded, which put intestine cells in rings I through IV closest to the microscope objective. Doing this avoids quantification error due to loss of signal with depth of tissue (i.e., imaging intestine through the germline when

animals orient on their right sides). The muscle cells were on the oblique, dorsal and ventral sides of the animals, and less easy to observe in the lateral orientation that animals tend to assume on slides and in these devices.

To image the animals, we used a 40×1.2 NA water objective on a Zeiss LSM780 confocal microscope. We excited the sample with 488 and 561 nm lasers and collected light from 490 to 550 nm for mEGFP signal, and from 580 to 640 nm for mCherry signal. We also collected transmitted light signal for Nomarski DIC images to aid in cell identification as needed. We focused on the same field of view for each animal- starting from the posterior of the pharynx to the first half of cells in intestinal ring IV. We collected images of the entire z depth of each animal, from one side to the other, using two micrometer step size and a two micrometer optical[34]. Additional information is available in ref. [34].

**Image cytometry.** Our image cytometry consists of manual cell identification and annotation, with a semiautomatic quantification step. Briefly, we first determined the orientation of the animals in images and then identified individual intestine or muscle cells. We then measured signal within an equatorial slice of the cell's nucleus, as a proxy for the whole cell, shown in Fig. [1]. Nuclear signal of freely diffusing monomeric fluorescent protein is nearly perfectly correlated with the cytoplasmic contents[34]. We used the ImageJ software (ImageJ version 1.53c) as well as custom built Nuclear Quantification Support Plugin called C. Entmoot (Alexander Seewald, Seewald Solutions, Inc., Vienna) for nucleus segmentation and signal quantification[35]. Additional image cytometry information is available in ref. [34].

**Data processing and noise calculations.** Here, we measured intrinsic noise by measuring the expression level of differently colored reporter alleles in two-day old adult animals that appear to be in a steady-state of gene expression[35]. Intrinsic noise is essentially the quantitative measure of relative deviation from the 1:1 ratio; data points having a 1:1 ratio fall on a 45° diagonal trend line. Intrinsic noise measures how deviant a pair of reporter alleles is from the average ratio among groups of cells, thus quantifying how probable it is to observe biased or monoallelic expression for a given gene (pair of alleles) in a given population of cells (e.g., muscle cells or intestine cells). The assumptions of our intrinsic noise model are the same as the assumptions in ref. [34]. We sometimes used 8-bit or 16-bit file settings during data collection, though this difference was obviated after normalization. We normalized expression level data for each allele to per-experiment means as in previous investigations[32,45,90]. We calculated intrinsic noise as detailed in refs. [32,45,90]. Specifically, the formula for calculating intrinsic noise is:

$$\text{Intrinsic noise} = ((x - y)^2)/(2\langle x \rangle \langle y \rangle)$$

where $x$ and $y$ are each cell's allele expression values and $\langle x \rangle$ and $\langle y \rangle$ are the average value for each allele. X,Y expression data and calculated noise values for each figure are available in Source Data, in Excel format. Numbers of cells and experiments are detailed in Supplementary Table 1.

**Plotting and statistics.** We used SigmaPlot 12.5 (Systat Software, Inc., San Jose) for all plotting and statistical analyses of intrinsic noise. All data was non-normally distributed, even after attempting log or natural log transformations, thereby requiring non-parametric statistics for analysis. We used Spearman's non-parametric rank order correlation for Spearman's coefficients of determination shown in Supplementary Table 1. For experiments with multiple groups analyzing hsp-90 alleles in intestine cells or different promoters in intestine cells, we ran ANOVA on Ranks followed by Dunn's pairwise comparisons. For all other experiments with only two groups, we ran a non-parametric Mann–Whitney U-test for each distinct set of experiments. Details of each test are shown in Supplementary Note 1.

**Reporting summary.** Further information on research design is available in the Nature Research Reporting Summary linked to this article.

## Data availability

The data that support this study are available from the corresponding author upon reasonable request. The dbMAE is at https://mae.hms.harvard.edu/. Source data are provided with this paper.

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

## Acknowledgements

We would like to thank Gary Ruvkun, Chris Link, George Martin, and Matt Kaeberlein for careful reading of manuscript drafts. We would like to thank Lu Wang, Theo Bammler, and James MacDonald at the University of Washington Nathan Shock Center for Excellence in the Basic Biology of Aging. Funding was provided by NIA R00AG045341 to A.M., and NCI R01CA219460 to A.M. and a Pilot Grant to A.M. from the University of Washington EDGE Center of the National Institutes of Health funded by NIEHS P30ES007033.

## Author contributions

A.M. and B.S. designed the study. B.S. and S.Y. performed experiments. B.S. and A.M. analyzed the data. B.S. and A.M. wrote the initial manuscript. B.S., S.Y., and A.M. revised the manuscript.

## Competing interests

The authors declare no competing interests.
