## [Peer Review File · Nature Communications]

Reviewers' Comments:

Reviewer #1:

Remarks to the Author:

The manuscript by Sands et al investigates control of noise in gene expression by introns using the *C. elegans* model system. The authors use allele specific reporters to reproduce the two colour-experiments of Elowitz. The results could be of potential interest. But, my major concern is about if the observed results is just the side-effect of intron control of mean expression. I have the following comments:

- In Figure 3H and 5C, what is exactly reported. There is no formula in the supplementary material. If this is intrinsic noise it is unites (it is not correct to say AU). Also, it is not clear what are samples that the box plot is plotted over (why are there so many estimates and why are there so much variability in the estimate of intrinsic noise?) In the supplementary page 16 they talk about intrinsic noise and intrinsic noise strength, are these different quantities? Why is the data scaled by 100?
- If the intrinsic noise is plotted against mean, how does this look like? Intrinsic noise goes down with the mean expression. Could the variation to the mean expression explain the observed changes in the intrinsic noise? If that is the case, then it is not really the noise that is regulated, but the observations are simply related to previously reported result on the effect of introns on mean expression.
- Could the authors comment on extrinsic noise and the total noise observed in the data. Some figure about this could be helpful. It seems that extrinsic noise maybe dominating? What are the possible sources of extrinsic noise.

Reviewer #2:

Remarks to the Author:

In this manuscript Sands et al. have explored the possibility that introns reduce stochastic allele expression bias during animal development. They have adapted a fluorescent reporter approach that has been developed in single cell organisms to measure differences in allele expression across different cell types and tissues in adult worms. The topic is certainly of a general interest and the effect of introns on allele specific gene expression variation is novel. However, because only single-copy transgenes have been used, I have some concerns about the methodological approach, the interpretation of some results, and the lack of validation of the stochastic allele expression variation using other methods.

Specific comments:

- The authors have used two different fluorescent single-copy transgenes (eGFP and mCherry) to evaluate allele expression variability in cell types and tissue. However, some variability in fluorescent intensity might be caused by differential protein stability between eGFP and mCherry in different cellular context, and from differences in autofluorescence in some specific cells or tissues. To address this concern, the authors should validate their findings using RNA smFISH to detect eGFP and mCherry transcripts. Another potential issue is the fact that single-copy transgene are subjected to stochastic somatic RNAi and/or chromatin silencing (Frøkjær-Jensen et al., Cell 2016). Different sequences (such as GFP and mCherry) might differentially activate the somatic silencing response in somatic cells and cause transgene expression variegation effect. Also, it has been shown that intronic sequences help to protect from silencing (Akay et al., Dev Cell 2017) and therefore a stochastic silencing effect might also explain why introns less genes show more variations.
- It is not clear from the representative images shown how the signal of eGFP or mCherry is specifically quantified at cellular levels since the proteins are diffuse everywhere and they might

overlap with other cells or tissue. Perhaps a nuclear localized eGFP and mCherry might help to quantify the signal in specific cells. In addition, as mentioned above the usage of an alternative method such as smFISH might help validate the findings.

- In Frøkjær-Jensen et al., Cell 2016 they have shown that single-copy transgenes are subjected to somatic variegation expression based on the chromosomal location of the transgene. At this purpose, maybe the authors should try to generate similar single-copy transgene using a different locus and test whether they observe different effects. Alternatively, they can use CRISPR-Cas9 to tag and monitor endogenous genes.

- One of the conclusions of the manuscript is that gene without introns might be subjected to a higher degree of variability. This hypothesis should be tested by monitoring allele expression on endogenous introns less genes using CRISPR-Cas9 tagging approach combined with RNA smFISH and microscopy measurement of protein expression.

- In all the scatter plots presented is not clear what the dots represents. They should specify in the legend whether the dots correspond to quantified signal in individual cells or tissue or whole worms as well as the number of worms used for the quantifications. Also, they should add on the graph some statistics and not only black and white distributed dots.

Reviewer #3:

Remarks to the Author:

Sand et al elegantly use the advantage of the nematode *C. elegans* to test if the presence of intron controls the stochasticity of autosomal allele expression for a given gene.

The authors use two different fluorescent reporters (EGFP and mCherry) with or without introns under the same promoter hsp-90 to measure the stochasticity of allele expression. In order to do that, they have engineered *C. elegans* genome and inserted at the same chromosome II locus one copy of either the EGFP or the mCherry reporter. They subsequently generated heterozygotes with one copy of the green fluorescent protein and one copy of the red fluorescent protein. Then they assessed for the expression of each allele in the different cells of the quantified tissue. Using hsp-90 promoter, they compared allele expression in the presence or not of introns in the intestine cells, but also in the muscle cells. Then to determine if the promoter sequence could affect allele expression, two additional promoters were used. They also compared if the intron position had an effect.

They concluded that indeed introns control stochastic autosomal biases using a 5'- position dependent mechanism.

MAJOR POINTS

1) Only one locus has been tested

It is known that the genetic environment could affect gene expression. Here, all the reporters have been inserted in the same locus on chromosome II.

An important control would be to perform the experiment with hsp-90 with or without intron on another chromosome. With the MosCI technique, single copy could be inserted at various loci.

2) Choice of the two additional promoters

The authors are using for most of their experiment the hsp-90 promoter. A general description of where this promoter is driving expression would have been nice.

In order to prove that their observations are not promoter based, Sand et al have used two other promoters, hsp-16.2 and vit-2 which are particular cases.

hsp-16.2 is a promoter that drive ubiquitous expression after a heat shock: on one hand, there is no indication in the Material and Methods on how they proceed to induce the expression of the reporter.

On the other hand, heat shocking the worms could affect gene regulation.

Moreover, vit-2 drives expression in the intestine, a polyploid tissue.

A third control such as the use of a promoter driving expression in the muscle (a diploid tissue) such

as myo-3 is missing. Similar strains as vit-2 could be generated and quantified.

MINOR POINTS

- 1) 2nd paragraph: It is not clear that the reporter with introns is used.
- 2) Fig2 legend: the information that we are studying reporter alleles with introns under the hsp-90 promoter is missing (even though it is mention in Fig1)
- 3) Fig2: orientation is missing (Dorsal, Ventral, left, right or Anterior, Posterior); could be added on the pictures.
- 4) Fig2: Reformat the scale bars.
- 5) Fig 3 C and D and Fig 4: need to add a scale bar.
- 6) Fig3H: Y axis legend is missing.
- 7) I couldn't find the formula on how the intrinsic noise is calculated ; it would be nice to have it on one of the figure (and not hidden in the text).

If the two controls were added, it would prove that *C. elegans* is a good model to study intron control of allele expression. If the two controls were added to the study with an additional effort on the esthetic of the figures, this paper would be a good candidate for Nature Communications.

We thank the reviewers for their comments and experimental suggestions. We believe the implementations of the reviewers' considerations and suggestions have significantly improved the scientific strength and clarity of the manuscript. We performed additional experiments and more thoroughly explained the current state of research on, and scientific understanding of, allele bias in the introduction.

In the interim period between the initial submission and this revision, in addition to the case reports of mutations in introns resulting in monoallelic expression we cited, new work analyzing human single cell RNA-seq data found that SNPs in introns are significantly enriched in instances of allele bias, compared to SNPs in coding sequences or 3'UTRs. This new human cell work on introns and allele bias adds weight to the significance of our findings because 1) this recent report further establishes the importance of understanding how introns influence allele bias by demonstrating broad effects in human cells, and 2) our experimental data discriminates between the possibilities that SNPs in the middle of introns alter splicing versus altering a cis element. Our experimental data shows that when we remove introns, bias increases. This is consistent with the idea that intron-SNP-associated allele bias is caused by loss of a cis element that prevented stochastic silencing, because there is no splicing in intronless alleles. Thus, the bias cannot be due to nonsense mediate decay of unspliced transcripts. We believe the resulting manuscript is now ready for publication.

Reviewer #1 (Remarks to the Author):

The manuscript by Sands et al investigates control of noise in gene expression by introns using the *C. elegans* model system. The authors use allele specific reporters to reproduce the two colour-experiments of Elowitz. The results could be of potential interest. But, my major concern is about if the observed results is just the side-effect of intron control of mean expression. I have the following comments:

We are grateful for the reviewer's comparison of our work to that of Elowitz. We believe that the reviewer is referring to Elowitz's (and Swain's) initial observations on noise in gene expression suggesting that expression level drives monoallelic expression and the observation that introns increase expression levels. More recent studies have found that allele expression bias in eukaryotes is not driven by expression level. For biased or monoallelically expressed genes, scientists have observed epigenetic modifications to chromatin consistent with changes in silencing and activating marks leading to the general consensus that histone-marking-driven transcriptional gene silencing causes stochastic allele bias/monoallelic expression. And, this is observed to happen across the expression range.

Relatively recent reports we now cite have shown that allele expression bias happens for genes at all expression levels, in yeast (2009) and in mammals (2007, 2013, 2015). Expression level does not dictate bias in eukaryotes. For example, in diploid yeast, when expression from all four strands of DNA was quantified on tiling arrays, extreme allele bias was detected in genes that spanned the entire expression level range. Bias spanned the range of expression in the 2007 study finding monoallelic expression to be relatively widespread in the thousands of genes they examined (around 10%ish), and it was not restricted to low expression. In mammals, only the top half of expressed genes were examined in ChIP seq studies successfully identifying monoallelic gene expression, and these relatively highly expressed genes would not be expected to be expressed in a

monoallelic or extremely biased fashion if expression level dictated bias. Yet, in metazoan cells, highly expressed genes like HSP90 are expressed in a monoallelic fashion. Here, we were able to study extreme allele bias *in vivo* using fluorescent reporter alleles. And, what we learned may actually provide insight into how introns can control mean expression. That is, part of the effect on mean expression may be the expression of both alleles.

In conclusion, in eukaryotes, allele expression bias is not due to low trans activating factors binding to only or mostly one allele of a gene, as it was shown to be in bacteria (which do not really have alleles like eukaryotes), and was thought to be in microbes (yeast). Monoallelic expression/stochastic allele bias is driven by an active silencing process – by whether or not an allele gets randomly silenced, and this happens for genes at all expression levels. So, part of the mechanisms by which introns increase expression levels may be through the prevention of stochastic somatic allele silencing, in addition to other potential mechanisms, none of which are well resolved yet, reviewed by Orit Shaul in 2016, which we now cite. This active, stochastic silencing process is presumably tuned over evolutionary time.

- In Figure 3H and 5C, what is exactly reported. There is no formula in the supplementary material. If this is intrinsic noise it is unites (it is not correct to say AU). Also, it is not clear what are samples that the box plot is plotted over (why are there so many estimates and why are there so much variability in the estimate of intrinsic noise?) In the supplementary page 16 they talk about intrinsic noise and intrinsic noise strength, are these different quantities? Why is the data scaled by 100?

We thank the reviewer for this observation. We previously only referenced the formula and we now include it in the experimental schematic Fig. 1, and in the materials and methods. We now label the y axis of our box plots as intrinsic noise. Regarding the various estimates, each individual cell gives a value for how deviated it is from a perfect 1:1 ratio. Because each cell gives an estimate, and we measured hundreds of cells, there are hundreds of data points in each box plot. Numbers of experiments and cells are now detailed in Table 2. We do not include noise strength as that was a theoretical assertion from Raser and O’Shea. We now do not mention or report intrinsic noise strength, and apologize for the previous artifactual inclusion. The data is no longer arbitrarily scaled as it was before to aid incremental comparisons for a previous reviewer.

- If the intrinsic noise is plotted against mean, how does this look like? Intrinsic noise goes down with the mean expression. Could the variation to the mean expression explain the observed changes in the intrinsic noise? If that is the case, then it is not really the noise that is regulated, but the observations are simply related to previously reported result on the effect of introns on mean expression.

We thank the reviewer for their careful examination of our work. We note above that significant body of evidence to date suggests that the mode of regulation for eukaryotic stochastic allele bias is stochastic silencing of alleles, and not low trans activating factors. We think the reviewer may be referring to the microbial trend of low expression and high noise in bacteria from Elowitz and Swain. This is not the case for genes in eukaryotes, as extreme allele bias is shown across expression level ranges, as we detailed above. As the reviewer states, intrinsic noise is unitless, making it difficult to

plot versus the mean. But, it might be that the reviewer wants to know if instances of low expression result in bias for a particular gene, or, if when comparing genes, genes with lower expression levels have higher bias/intrinsic noise. We can see the former in scatter plots rather easily, but for the latter point, we list the large, qualitatively observable differences in expression of the fluorescent alleles we examined in relation to their noise. We do not feel this is important to include in the manuscript as noise is not dictated by expression level in metazoans.

Allele bias is seen at all expression levels in muscle cells. Expression level for a given allele for a particular gene does not dictate whether an allele will be expressed in a monoallelic fashion or not. Stochastic allele bias happens for alleles with or without introns – just less frequently for alleles with introns. Figure 2a,c shows that some virtual monoallelic expression occurs when alleles are expressed at a high level for *hsp-90* in muscle cells. So, bias does not seem to be caused by weak expression.

Additionally, between genes, if this is what the reviewer is referring to, the idea that low expression change causes increases in bias is not true with our data either. While we can easily work with *hsp-90* transcriptional reporters on stereoscopes, we could barely detect *myo-3* or *MTL-2* alleles, because they are so dimly expressed. Yet, these dimly expressed genes are just as biallelically expressed as high, intestinally expressed *vit-2*, shown in Table 2. And besides our data, there is the data from yeast tiling arrays and mammalian cells showing bias across the entire expression range. Stochastic allele bias/monoallelic expression is believed to be caused by stochastic silencing, which reviewer 2 also provides useful references for we now cite. Our data is consistent with cis elements present in introns preventing silencing causing bias.

Finally, we also note that not all introns affected bias. The *MTL-2* alleles using the T2A peptide did not have a significant or qualitative difference in allele bias when comparing the genes with or without the single natural 5'-intron in *mtl-2*. We also know from other work that not all introns affect expression level either. So these results are not unexpected, and may inform position/sequence requirements for strong effects of introns on bias.

- Could the authors comment on extrinsic noise and the total noise observed in the data. Some figure about this could be helpful. It seems that extrinsic noise maybe dominating? What are the possible sources of extrinsic noise.

We thank the reviewer for this careful observation and refer them to Burnaevskiy et al 2019 [10.1038/s41467-019-13664-7](https://doi.org/10.1038/s41467-019-13664-7) and Colman-Lerner et al 2005 [10.1038/nature03998](https://doi.org/10.1038/nature03998). This extrinsic noise is now known to be gene expression capacity and it has to do with cells and animals ability to turn genes into protein, and it is extensively discussed in those two manuscripts. Additional studies focusing on gene expression capacity in the context of its consequences are also available for mammalian cells, like Spencer et al 2009, focusing on HeLa cells and apoptosis. Additional references for mammalian papers examining this are cited in our recent review in Geroscience on the mechanisms of cell to cell variation in gene expression and aging: <https://doi.org/10.1007/s11357-021-00339-9>. We now refer to our findings on extrinsic noise in *C. elegans*, aka protein expression capacity in Burnaevskiy et al 2019, in the first section of results in the revised manuscript.

Reviewer #2 (Remarks to the Author):

In this manuscript Sands et al. have explored the possibility that introns reduce stochastic allele expression bias during animal development. They have adapted a fluorescent reporter approach that has been developed in single cell organisms to measure differences in allele expression across different cell types and tissues in adult worms. The topic is certainly of a general interest and the effect of introns on allele specific gene expression variation is novel. However, because only single-copy transgenes have been used, I have some concerns about the methodological approach, the interpretation of some results, and the lack of validation of the stochastic allele expression variation using other methods.

We thank the reviewer for such detailed comments. We have generated 16 new strains of worms in order to test our methods at multiple loci, with additional tissue-specific promoters, and more importantly, on natural *hsp-90* and *mtl-2* gene sequences. To accomplish the later, we used a T2A strategy where we removed the native introns from *hsp-90* and *mtl-2*. We have added the additional experimental details to Figure 1. Importantly, the data for our new *hsp-90* T2A constructs corroborates our prior results with *hsp-90* transcriptional reporter alleles.

Specific comments: - The authors have used two different fluorescent single-copy transgenes (eGFP and mCherry) to evaluate allele expression variability in cell types and tissue. However, some variability in fluorescent intensity might be caused by differential protein stability between eGFP and mCherry in different cellular context, and from differences in autofluorescence in some specific cells or tissues. To address this concern, the authors should validate their findings using RNA smFISH to detect eGFP and mCherry transcripts.

mCherry and mEGFP do not have differential stability in somatic cells. We submit to the reviewers our results from Sands et al. 2018, [doi: 10.1016/j.tma.2018.01.001](https://doi.org/10.1016/j.tma.2018.01.001). In this paper we expressed 10 different fluorescent proteins in *C. elegans* ranging across the spectrum, from mTagBFP-2 (456nm), to mNeptune (634nm), and including mEGFP and mCherry. We did not observe any difference in localization or stability of fluorescent proteins. Patterns of expression on chromosomes I, II and V were identical. Moreover, if there were stability differences, we would never be able to detect changes in allele bias within a tissue, and we would always see one tissue as more or less biased for a fluorescent protein. Most cells in our experimental animals are in a state of near perfect biallelic expression.

Consistent with the above assertions, our results in intestine and muscle cells show we are able to detect both high and low allele bias with reporter alleles. This would not be possible with systemic bias caused by mCherry or GFP sequences in the somatic cells we measured. We now show that we can detect both low and high stochastic allele bias in *both* muscle and intestine cells using mEGFP and mCherry alleles, shown with new experiments with *myo-3* in muscle cells in Fig. 2. We are also able to detect high and low noise in intestine cells. So differences in bias between tissues are not intrinsic to the tissue, otherwise we would not have been able to detect

low allele bias differences attributable to *myo-3* (relative to *hsp-90*), which is expressed exclusively in muscle.

Regarding autofluorescence and smFISH, we believe our methods intrinsically address concerns of signal contamination. Because we use a spectral confocal microscope to measure fluorescent protein signal from inside the equatorial plane of each 2µm thick optically sliced nucleus z section, there is no concern about autofluorescence from secondary lysosomes, because there are none in the nucleus. Thus, there is no need to measure transcripts in the nucleus with smFISH, because we are already using naturally concentrated nuclear signal from expressed fluorescent protein. Unfortunately, in the previous version of our manuscript, we did not clearly state our approach, and the reviewer expressed understandable concern. We now more clearly describe our methods in Image Cytometry in Materials and Methods and in Figure 1 with an additional image. Confocal microscopy obviates signal pollution contribution from other subcellular sections or tissues in other optical planes in z. Our confocal image cytometry methods are proven to result in reproducible cell-specific quantification of signal, so that concern of signal pollution is inherently addressed with our image cytometry methods (Mendenhall et al 2015, Burnaevskiy et al 2019). And RNA for allele bias *in vivo* is not advisable with the current data from Lo and Chen 2019. As an additional approach, we now also use T2A peptides fused to native coding sequences to cause expression of native proteins with fluorescent proteins in 1:1 fashion, separating them at the ribosome.

One of the advantages of our approach is measuring expression at the protein level. Protein is what matters for most genes. Lo and Chen 2019 measured allele bias with transcripts and protein and found that the transcripts did not correlate with the protein abundance at any given point in time, likely due to transcriptional bursting during development. The measurements of transcripts may catch instances of transcriptional bursting *in vivo* and not actual bias at the protein level, as seems to have been the case in Lo and Chen 2019. Cell culture studies are able to study bias at the transcript level because, for many genes the allele bias becomes fixed and mitotically stable in culture (See Chess et al 2016, and controversial discussion of bias in 2018 Nat Gen article we cite in the intro).

Another potential issue is the fact that single-copy transgene are subjected to stochastic somatic RNAi and/or chromatin silencing (Frøkjær-Jensen et al., Cell 2016). Different sequences (such as GFP and mCherry) might differentially activate the somatic silencing response in somatic cells and cause transgene expression variegation effect.

We thank the reviewer for their comments and attention to the details of this work. The stochastic silencing the reviewer refers to happens at relatively rare, non-permissive loci, near the ends of chromosomes. We have revised the manuscript to show that we used expression permissive loci that are not known to be subject to stochastic somatic RNAi or chromatin silencing.

Regarding concern of the use of single copy transgenes: The transgenes we used here are not systematically silenced in a heritable fashion as has been observed for germline expression of transgenes in the Akay et al paper the reviewer cites. Nor are they inherently variegated at the loci we used, as we used validated, expression-permissive loci, and not nonpermissive loci at the ends of chromosomes reported by C.F.J and colleagues in the Cell 2016 paper the reviewer cites. In that paper miniMos insertions

were silenced for both alleles in most cells when expressed from loci near the ends of chromosomes, shown in Fig. 2 (they did not quantify distinctly colored alleles). In fact, these variegated loci were the vast minority of the hundreds of loci they surveyed. In this report, we used autosomal, somatic and germline expression permissive loci validated by CFJ.

If different sequences of mCherry and mEGFP (XFPs) differentially activated the somatic silencing response, we would only detect systemic bias or incoherent noise, and not differences in bias attributable to promoters, loci and cell types. Expression patterns would never be detectable, but they are, as in Burnaevskiy et al 2019 or Sands et al 2018, where the same expression patterns are seen for *eft-3*-promoter-driven reporters at multiple expression permissive loci. For example, we can see near perfect correlation between these fluorescent proteins, with or without introns, for *vit-2* and *myo-3* driven alleles. And we can see nearly zero correlation between these same two fluorescent proteins, with or without introns, for *hsp-90* alleles on chromosome V, compared to the same alleles expressed from Chromosome II (both permissive sites) in muscle or intestine cells, detailed in Table 2. So, we are able to detect a range of effects, *consistently, across many experiments*, with these XFP coding sequences, demonstrating that allele bias is not inherently caused or prevented by these XFP sequences. Our data show that promoters, loci and introns are controlling the amount of allele bias. If we were not able to detect those effects, then we would assume that there was some inherent instability, but that is not the case. Furthermore, we saw the same effects with full length gene coding sequences for *hsp-90* with or without its natural introns in our T2A experiments.

In this report, we examined somatically expressed genes that have been expressed in the animals for 5-30 generations before being measured for experiments. Thus, our experimental animals are not subject to germline silencing because they are somatically expressed, nor are they subject to variegation stemming from nonpermissive site location because we used permissive sites. We do note that single copy transgenes can be silenced entirely, but they are silenced upon generation and reversed upon *csr-1(RNAi)*, shown by Craig Mello's group.

To be clear, our work differs from Akay and other work by CFJ because we are looking solely in somatic cells in day 2 adults and not in the germline or in larvae, and because we are inserting our experimental genes at expression-permissive loci, as validated by CFJ and colleagues.

We note that Frøkjær-Jensen, et al Cell 2016 did do a large amount of work characterizing gene expression as a function of locus, expanding on a previous report from 2014, with more loci and more advanced automated characterization of L1 animals' gene expression levels at cell resolution. We now cite them for their work on introns and position effects on expression patterns. Our focus is on somatic stochastic allele expression bias that often manifests as partial expression of one allele and high expression of another, shown in our scatter plots. We believe the revised introduction and Figure 1 now more clearly state these facts and thank the reviewer for the help clarifying these concepts by more precisely framing them in their relevant biological contexts.

Also, it has been shown that intronic sequences help to protect from silencing (Akay et al., Dev Cell 2017) and therefore a stochastic silencing effect might also explain why introns less genes show more variations.

We thank the reviewer for these insights and opportunities to clarify our approach and experimental system. As silencing is the major cause of stochastic allele bias in metazoans and seemingly single celled eukaryotes as well, we agree that introns are likely preventing stochastic, cell autonomous allele silencing in somatic cells and have revised the manuscript accordingly to mention this in the introduction and discussion. We note that allele expression bias and silencing are likely distinctly regulated in the soma and in the germline, with germline expression being the focus of the above cited papers.

- It is not clear from the representative images shown how the signal of eGFP or mCherry is specifically quantified at cellular levels since the proteins are diffuse everywhere and they might overlap with other cells or tissue. Perhaps a nuclear localized eGFP and mCherry might help to quantify the signal in specific cells. In addition, as mentioned above the usage of an alternative method such as smFISH might help validate the findings.

We thank the reviewer for this observation, which has provided us the opportunity to more clearly describe our methods in the now revised Figure 1. In fact, we do analyze eGFP and mCherry signal from the nucleus but we did not make that important fact clear in the prior submission. The signal from the nucleus is visibly concentrated due to a slower diffusion coefficient in the nucleus, which we reported in Mendenhall et al 2015, corroborating other results in yeast and mammalian cell culture.

The methods for quantifying signals from individual cells were previously detailed deep in the Image Cytometry part of the Methods, and are now shown graphically in Figure 1, with additional details in Methods.

Finally, smFISH suffers from the same out of plane fluorescence as fluorescent protein, and mRNA does not often correlate with protein level. smFISH cannot discriminate between allele bias and allele specific bursting. mRNA measures for studying allele bias in cell culture can be sufficient because allele bias in cultured cells can be stable at the RNA level for years. However, in animals allele specific bursting of transcription can muddle what will or will not be allele bias at the protein level, recently shown in drosophila by Lo and Chen 2019, which we now reference. As stated above, the advantage of measuring at the protein level allows us to quantify expressed protein, and is part of the technical advance of our approach. Thus, to address reviewer concerns we took another approach of using full length natural genes, observed with a diffusible fluorescent protein that is made each time a transcript is translated, via the addition of T2A sequences linking fluorescent protein sequences to natural coding sequences, thereby permitting the observation of expression of individual alleles.

- In Frøkjær-Jensen et al., Cell 2016 they have shown that single-copy transgenes are subjected to somatic variegation expression based on the chromosomal location of the transgene. At this purpose, maybe the authors should try to generate similar single-copy transgene using a different locus and test whether they observe different effects. Alternatively, they can use CRISPR-Cas9 to tag and monitor endogenous genes.

We thank the reviewer for these suggestions. We have generated 16 new worm strains to test these suggestions. The new strains include identical *hsp-90* reporter alleles moved from Ch II to Ch V. They also include full length HSP-90::T2A and MTL-2::T2A constructs which allowed us to assess the role of native introns in the context of native genes, without

disrupting endogenous *hsp-90* and *mtl-2*. To assess intron effects on allele bias in these animals, we removed the native introns from the natural gene sequences.

We note that in that cited paper, the authors showed that stochastic silencing happens at relatively rare nonpermissive loci near the ends of chromosomes. Now, we more clearly explain that we only measured expression at permissive loci, which CFJ actually validated, and we now detail in Fig. 1. We also note that at the permissive loci we examined on chromosomes I, II and V, we never observed whole animal, biallelic silencing in entire tissues, like at nonpermissive loci in CFJ 2016.

- One of the conclusions of the manuscript is that gene without introns might be subjected to a higher degree of variability. This hypothesis should be tested by monitoring allele expression on endogenous introns less genes using CRISPR-Cas9 tagging approach combined with RNA smFISH and microscopy measurement of protein expression.

We thank the reviewer for this insightful suggestion. We have done our best to answer it through a couple of additional targeted approaches. Rather than an ad hoc tagging approach, which can be misleading for reasons we detail below, we performed bioinformatic analyses and removed natural introns from natural genes.

To robustly test the question, “do intronless genes show more allele bias/variability”, we bioinformatically asked the question “are intronless genes enriched for monoallelic expression for all intronless genes in the human genome?”. Additionally, we experimentally removed introns from the coding sequence of natural genes, *hsp-90* and *mtl-2*, using a T2A approach detailed in Figure 1.

While we would love to tag and survey allele bias for intronless genes, we cannot expect that an intronless gene would be noisier than *any* intron bearing gene, because each gene has its own setpoint, dependent on locus and tissue and promoter. That is, we believe an *ad hoc* approach to testing just a few intronless genes in *C. elegans* would be potentially misleading because these other factors also contribute to the stochastic allele bias. For example, a gene that has evolved to be intronless, as many may have because of intron effects on bias, may still be noisier/more biased than it was, but it might not be noisier than an intron bearing gene at a noisy locus. And that is why we took the bioinformatic approach for testing if human intronless genes were enriched at the monoallelic gene expression database, which has cataloged monoallelic expression for all human and mouse genes.

We now more clearly explain how we performed a bioinformatic analysis of human intronless genes cross-referenced to human monoallelic gene expression database (dbMAE), finding that there is statistically significant enrichment for monoallelic expression for the intronless genes.

- In all the scatter plots presented is not clear what the dots represents. They should specify in the legend whether the dots correspond to quantified signal in individual cells or tissue or whole worms as well as the number of worms used for the quantifications. Also, they should add on the graph some statistics and not only black and white distributed dots.

We thank the reviewer for this suggestion and implemented it in our figure legends. We measured single cells. We now clearly state that each dot is a single cell and have

designed an experimental schematic figure (Fig. 1) to more clearly demonstrate that point. We now show the box plots of intrinsic noise next to the scatter plots. We present results of statistical analyses in Table 2 and next to each result in the text and detail statistical analyses in the Methods and explicitly in the Supplemental Material.

Reviewer #3 (Remarks to the Author):

Sand et al elegantly use the advantage of the nematode *C. elegans* to test if the presence of intron controls the stochasticity of autosomal allele expression for a given gene. The authors use two different fluorescent reporters (EGFP and mCherry) with or without introns under the same promoter *hsp-90* to measure the stochasticity of allele expression. In order to do that, they have engineered *C. elegans* genome and inserted at the same chromosome II locus one copy of either the EGFP or the mCherry reporter. They subsequently generated heterozygotes with one copy of the green fluorescent protein and one copy of the red fluorescent protein. Then they assessed for the expression of each allele in the different cells of the quantified tissue. Using *hsp-90* promoter, they compared allele expression in the presence or not of introns in the intestine cells, but also in the muscle cells. Then to determine if the promoter sequence could affect allele expression, two additional promoters were used. They also compared if the intron position had an effect. They concluded that indeed introns control stochastic autosomal biases using a 5'-position dependent mechanism.

MAJOR POINTS

- 1) Only one locus has been tested

It is known that the genetic environment could affect gene expression. Here, all the reporters have been inserted in the same locus on chromosome II.

An important control would be to perform the experiment with *hsp-90* with or without intron on another chromosome. With the MosCI technique, single copy could be inserted at various loci.

We thank the reviewer. We have tested two additional loci now. We specifically tested *hsp-90* at two distinct loci on chromosomes II and V, allowing us to directly test the effects of locus on allele bias. We believe this is the first time that the exact same set of alleles has been moved to a different chromatin environment and examined at the level of the allele. We found that expression on chromosome V was indeed noisier, yet we could still detect an intron effect on allele bias.

2) Choice of the two additional promoters

The authors are using for most of their experiment the *hsp-90* promoter. A general description of where this promoter is driving expression would have been nice. In order to prove that their observations are not promoter based, Sand et al have used two other promoters, *hsp-16.2* and *vit-2* which are particular cases. *hsp-16.2* is a promoter that drive ubiquitous expression after a heat shock: on one hand, there is no indication in the Material and Methods on how they proceed to induce the expression of the reporter. On the other hand, heat shocking the worms could affect gene regulation.

Moreover, *vit-2* drives expression in the intestine, a polyploid tissue.

A third control such as the use of a promoter driving expression in the muscle (a diploid tissue) such as *myo-3* is missing. Similar strains as *vit-2* could be generated and quantified.

We now detail that *hsp-90* drives ubiquitous expression in the first section of the results and in Figure S3. We now detail how we perform heat shock to induce *hsp-16.2* expression in the methods. We believe showing introns still have their effect under diverse regulation, including after heat shock only strengthens our results by more robustly testing the hypothesis. We have now generated *myo-3* reporter alleles at the chromosome I locus using CRISPR/SKILodge system. We now clearly quantify stochastic allele bias in two distinct tissues, the diploid striated muscles and the polyploid intestine cells. Interestingly, both tissue-specific promoters were not very noisy, though they are expressed at vastly different levels, with *myo-3* being relatively hard to detect and *vit-2* being among the most highly expressed genes in the nematode.

MINOR POINTS

1) 2nd paragraph: It is not clear that the reporter with introns is used.

We now state that we surveyed the somatic tissues with *hsp-90* reporter alleles with introns and show this in figure S3.

2) Fig2 legend: the information that we are studying reporter alleles with introns under the *hsp-90* promoter is missing (even though it is mention in Fig1)

We now state that we surveyed the somatic tissues with *hsp-90* reporter alleles with introns and show this in figure 1.

3) Fig2: orientation is missing (Dorsal, Ventral, left, right or Anterior, Posterior); could be added on the pictures.

Corrected. We added Dorsal and Anterior designations.

4) Fig2: Reformat the scale bars.

Corrected. We removed the artifactually included small red scale bars from our microscopy program and moved all scale bars to the bottom right of each image.

5) Fig 3 C and D and Fig 4: need to add a scale bar.

Corrected, and now renamed as Figs. 3&5.

6) Fig3H: Y axis legend is missing.

Corrected.

7) I couldn't find the formula on how the intrinsic noise is calculated ; it would be nice to have it on one of the figure (and not hidden in the text).

Corrected. We added it to the experimental schematic and explicitly in the methods text.

If the two controls were added, it would prove that *C. elegans* is a good model to study intron control of allele expression. If the two controls were added to the study with an additional effort on the esthetic of the figures, this paper would be a good candidate for Nature Communications.

We added the extra locus and the extra promoter and two natural coding sequences and thank the reviewer for suggesting additional experiments that now more robustly support these findings.

Reviewers' Comments:

Reviewer #1:

Remarks to the Author:

The authors have addressed all my comments and have fully revised the paper with additional data and analysis. I am happy for the paper to get accepted as it is.

Reviewer #2:

Remarks to the Author:

In this revised version, the authors have addressed most of my concerns. They have generated new strains having the reporter on different chromosomal loci as well as modified endogenous genes to test whether the lack of introns results in increased stochastic allele expression. They have also clarified the methodology used in the manuscript. Overall, I think the paper can be published now on Nature communications.

Reviewer #3:

Remarks to the Author:

I think the authors have addressed all the comments in a reasonable way and have also included new data and reformat the paper that improves the manuscript.

I have just a few suggestions for the sake of clarity in the discussion:

- 1) The authors did not detect a difference in *mtl-2* allele expression bias: one provided explanation is that the allele bias could change after induction of expression caused by a stressor; my concern is that they have also used the heat-shock promoter *hsp-16.2* that is induced only after a heat shock which is also a stressor and this point is not addressed.
- 2) the last part of the discussion related to "Stochastic Allele Bias, Introns and Human disease" is really convincing except for the first example. Admittedly, the case of *PIT1* is an example of MAE but I do not see how it is related to the effect of intron in MAE.
- 3) In the third paragraph of this section, the authors mention the case of the *COL6A2*-associated Bethlem myopathy. I find that the sentence "The patient's unaffected parents had the same two alleles." is not clear. I would re-write it because if I understand correctly one parent is having the mutation within the intron and the other one is having the mutation in exon 28.
- 4) To mimic cases found in human, it would have been of interest to look at heterozygote strains such as *hsp-90::gfp w/introns// hsp-90::mcherry no introns* and vice versa to check if the fluorescent protein without introns is silenced.

Supp Fig 1; an effort has to be made: all the scale bars should be the same width and micrometers should be abbreviated; Supp fig 1b and 1f should be oriented the same way as Supp fig 1g (it is a convention); moreover, the authors should indicate the orientation with a schematic cross indicating AP axis and DV axis.

Supp Fig 2 is not necessary: it is redundant with Fig 1 and Fig 8.

Supplementary Table 1: EGFP should be lower case because it refers to the gene and not to the protein; same as for *mcherry*, *hsp-90* and *mtl-2*. More over the hyphen for *hsp-90/HSP-90* and *mtl-2/MTL-2* should not be forgotten.

AUTHORS' RESPONSE TO REVIEWERS' COMMENTS

Our point by point response to reviewers' comments is shown in red, below.

REVIEWERS' COMMENTS

Reviewer #1 (Remarks to the Author):

The authors have addressed all my comments and have fully revised the paper with additional data and analysis. I am happy for the paper to get accepted as it is.

We thank Reviewer #1 for their review of our manuscript.

Reviewer #2 (Remarks to the Author):

In this revised version, the authors have addressed most of my concerns. They have generated new strains having the reporter on different chromosomal loci as well as modified endogenous genes to test whether the lack of introns results in increased stochastic allele expression. They have also clarified the methodology used in the manuscript. Overall, I think the paper can be published now on Nature communications.

We thank Reviewer #2 for the review of our manuscript.

Reviewer #3 (Remarks to the Author):

We thank Reviewer #3 for the review of our manuscript. We address each remaining concern below, and in the manuscript as suggested by the reviewer.

I think the authors have addressed all the comments in a reasonable way and have also included new data and reformat the paper that improves the manuscript.

I have just a few suggestions for the sake of clarity in the discussion:

1) The authors did not detect a difference in *mtl-2* allele expression bias: one provided explanation is that the allele bias could change after induction of expression caused by a stressor; my concern is that they have also used the heat-shock promoter *hsp-16.2* that is induced only after a heat shock which is also a stressor and this point is not addressed.

We now discuss that the allele bias and the effect of introns on allele bias may be distinct under conditions of heavy metal expression induction.

“Second, unlike, *hsp-16.2*, the *mtl-2* gene has a constitutive expression level that allowed us to observe it without exogenous induction. Thus, an intriguing possibility is that allele bias could

change in an intron-dependent fashion after induction of expression caused by a stressor, as MTL-2 is induced by exogenous heavy metals, like cadmium⁷⁵.”

2) the last part of the discussion related to “Stochastic Allele Bias, Introns and Human disease” is really convincing except for the first example. Admittedly, the case of PIT1 is an example of MAE but I do not see how it is related to the effect of intron in MAE.

We believe the reference to PIT1 is useful for highlighting the general role of monoallelic expression in non-Mendelian genetic diseases. We use this as one of the few clear clinical examples of monoallelic expression affecting patient health to introduce the topic of monoallelic expression affecting human health. Then we discuss the intron-specific cases, after using the reference to PIT1 to introduce the reader to the concept.

3) In the third paragraph of this section, the authors mention the case of the COL6A2-associated Bethlem myopathy. I find that the sentence “The patient’s unaffected parents had the same two alleles.” is not clear. I would re-write it because if I understand correctly one parent is having the mutation within the intron and the other one is having the mutation in exon 28.

Thank you for pointing out this unclear statement. We revised for clarity, removing unnecessary reference to parents.

4) To mimic cases found in human, it would have been of interest to look at heterozygote strains such as hsp-90::gfp w/introns// hsp-90::mcherry no introns and vice versa to check if the fluorescent protein without introns is silenced.

We agree and have added this to a list of experiments for subsequent studies. We believe that *C. elegans* can be a very informative model system for studying the mechanisms behind allele bias, and its consequences. Until now, in animals, allele bias has basically been studied at the RNA level by RNAseq and FISH (with some identification through ChIP-seq). The system we describe in this manuscript allows us to study allele bias in the tissues of live animals, more closely mimicking living human tissue. The experiments Reviewer #3 suggests fall in line with our research plans. We thank the reviewer for their suggestions, this one in particular.

Supp Fig 1; an effort has to be made: all the scale bars should be the same width and micrometers should be abbreviated; Supp fig 1b and 1f should be oriented the same way as Supp fig 1g (it is a convention): moreover, the authors should indicate the orientation with a schematic cross indicating AP axis and DV axis.

Revised according to the reviewer’s guidance.

Supp Fig 2 is not necessary: it is redundant with Fig 1 and Fig 8.

Supp Fig. 2 has been deleted as per the reviewer’s guidance.

Supplementary Table 1: EGFP should be lower case because it refers to the gene and not to the protein; same as for mcherry, hsp-90 and mtl-2. More over the hyphen for hsp-90/HSP-90 and mtl-2/MTL-2 should not be forgotten.

Corrected.